# A PPP-type pseudophosphatase is required for the maintenance of basal complex integrity in *Plasmodium falciparum*

Alexander A. Morano[1,2], Rachel M. Rudlaff[1,2] & Jeffrey D. Dvorin [2,3] ✉

During its asexual blood stage, *P. falciparum* replicates via schizogony, wherein dozens of daughter cells are formed within a single parent. The basal complex, a contractile ring that separates daughter cells, is critical for schizogony. In this study, we identify a *Plasmodium* basal complex protein essential for basal complex maintenance. Using multiple microscopy techniques, we demonstrate that PfPPP8 is required for uniform basal complex expansion and maintenance of its integrity. We characterize PfPPP8 as the founding member of a novel family of pseudophosphatases with homologs in other Apicomplexan parasites. By co-immunoprecipitation, we identify two additional new basal complex proteins. We characterize the unique temporal localizations of these new basal complex proteins (late-arriving) and of PfPPP8 (early-departing). In this work, we identify a novel basal complex protein, determine its specific role in segmentation, identify a new pseudo-phosphatase family, and establish that the *P. falciparum* basal complex is a dynamic structure.

Malaria is one of humanity's most ancient pathogens; so significant that hemoglobinopathies including thalassemias and sickle-cell trait persist because they provide protection from severe malaria[1–3]. In 2021, there were 240 million cases of malaria and about 627,000 deaths, mostly in children under five infected with *Plasmodium falciparum*[4]. Despite steady improvements, progress toward malaria eradication has slowed in recent years[4–6]. One significant hurdle toward malaria eradication is an incomplete understanding of fundamental parasite biology. *P. falciparum* is a member of the Apicomplexa, a divergent clade of obligate parasitic eukaryotes, and its genome bears little resemblance to model organisms with >50% of its genes lacking homology to known proteins[7,8]. Thus, highly conserved processes like cell division, are divergent in *Plasmodium*.

Asexual replication of *P. falciparum* in erythrocytes causes clinically relevant malaria symptoms. The parasite divides via schizogony, wherein the organellar contents for 20–36 daughter cells (merozoites) are generated within a common cytoplasm followed by segmentation, a partitioning of contents simultaneous with a synchronized cytokinesis[9–12]. Segmentation requires two multi-component protein

structures known as the basal complex (BC) and the inner membrane complex (IMC)[13–15]. The IMC is a double membrane structure of flattened vesicles and proteins underlying the plasma membrane, providing structure to forming merozoites and anchoring the actin-myosin motor during invasion[14,16–18]. In *P. falciparum* merozoites, the IMC is one continuous vesicle. A set of apical polar rings anchor the subpellicular microtubules[19–21] marking its apical end and the BC marks its basal end.

The BC is a contractile ring positioned at the IMC's leading edge during segmentation[15,22]. It contracts as segmentation concludes and is hypothesized to act as a docking site for cytoskeletal proteins joining the IMC and mediate abscission of daughter parasites after segmentation[15,23–26]. The only known *P. falciparum* BC proteins are PfCINCH, PfBLEB, PfBCP1, PfBTP1, PfBTP2, PfHAD2, and PfMORN1[15,22,27–30]. Of these, only PfCINCH, PfMORN1, and PfBLEB have been functionally investigated[15,28,29]. While PfMORN1 and PfBLEB are dispensable for asexual replication[15,28], PfCINCH *is* required for segmentation but its molecular role in BC function remains unknown[15].

[1]Biological and Biomedical Sciences, Harvard Medical School, Boston, MA 02115, USA. [2]Division of Infectious Diseases, Boston Children's Hospital, Boston, MA 02115, USA. [3]Department of Pediatrics, Harvard Medical School, Boston, MA 02115, USA. ✉e-mail: Jeffrey.Dvorin@childrens.harvard.edu

In the related Apicomplexan parasite, *Toxoplasma gondii*, the BC has been extensively studied and multiple proteins have been characterized[23–26,31,32]. In *T. gondii*, the class VI-like[33] myosin TgMyoJ and the centrin TgCen2 are hypothesized to mediate BC contraction[25,34]. Moreover, the *T. gondii* BC is dynamic,[25,35] with a well-constructed timeline of assembly and recruitment[31].

Here, we identify and characterize PfPPP8—an essential *Plasmodium falciparum* BC protein and novel pseudophosphatase—a protein bioinformatically characterized as a phosphatase but possessing a mutation inhibiting enzymatic activity. Pseudophosphatases compose 14% of identified phosphatases and play varied roles, including regulating active phosphatases by temporarily sequestering phosphor-protein substrates[36,37]. Pseudophosphatases have been found in nearly every phosphatase family, but PfPPP8 is the first identified within the phosphoprotein phosphatase (PPP) serine-threonine phosphatase family[36].

Using super-resolution, live cell, and expansion microscopy (Ultrastructure Expansion Microscopy (U-ExM)), we show that PfPPP8 is required to maintain BC integrity, but not for initial BC formation. We demonstrate the temporal localization of PfPPP8, which departs from the BC before contraction finishes, and identify and validate two additional BC proteins, PfMyoJ and PF3D7_0214700, recruited to the BC halfway through segmentation. This study establishes the *Plasmodium* BC as a dynamic structure and characterizes a novel, essential component.

## Results

### PfPPP8 is a novel, essential member of the *Plasmodium falciparum* basal complex

PF3D7_1018200/PfPPP8 was identified via immunoprecipitation with the BC protein PfCINCH[15]. It has a transcriptional profile similar to PfCINCH's[38], is predicted to be essential based on experiments in *P. berghei* and a transposon insertion screen in *P. falciparum*[39,40], and has a predicted serine-threonine phosphatase domain[41]. To evaluate the localization of PfPPP8, we fused the spaghetti monster-V5 (smV5)[42] epitope tag to the C-terminus of PfPPP8 in the 3D7 parasite strain (Supplementary Fig. 1a, b). By immunofluorescence, PfPPP8-smV5 colocalized with the BC proteins PfMORN1 and PfBCP1 (Fig. 1a), demonstrating PfPPP8 is a bona fide BC protein.

To determine if PfPPP8 was essential for asexual replication, we utilized the Tet repressor (TetR)-binding aptamer system[43]. In the PfPPP8-smV5 strain, we included ten copies of the TetR-binding aptamer in the 3' untranslated region and the TetR-DOZI fusion protein under the control of a constitutive *Plasmodium* promoter[43] (Supplementary Fig. 1a, b). In the presence of the TetR ligand, anhydrotetracycline (ATc), the TetR-DOZI fusion protein is released from the aptamer array and PfPPP8 is translated[43,44]. In the absence of ATc, TetR binds the Tet-aptamers and the fused PfDOZI sequesters target mRNA in stress granules preventing translation[43] (Supplementary Fig. 1c).

We performed a three-cycle growth curve of parasite growth with and without ATc. PfPPP8-smV5$^{Tet}$ parasites failed to replicate without ATc (Fig. 1b). In PfPPP8-sufficient (+ATc) parasites, parasitemia increased from $0.25 \pm 0.02\%$ initially to $8.11 \pm 0.19\%$ during cycle 3. In PfPPP8-deficient (-ATc) parasites, parasitemia increased from $0.28 \pm 0.02\%$ to only $0.35 \pm 0.04\%$ parasitemia during cycle 3 (Fig. 1b). This >95% replication failure corresponds to an $86 \pm 4\%$ PfPPP8 knockdown at the protein level (Fig. 1c and Supplementary Fig. 1d).

### PfPPP8 knockdown causes segmentation failure

When arrested with the cysteine protease inhibitor E64, parasite egress is blocked after rupture of the parasitophorous vacuolar membrane (PVM) but prior to rupture of the RBC plasma membrane[45–47]. In wild-type E64-stalled parasites, collections of merozoites are contained within the semi-intact RBC plasma membrane (Supplementary Fig. 2a). In contrast, E64-stalled PfPPP8-deficient schizonts formed abnormal merozoite agglomerates, suggesting a defect in merozoite formation

or separation (Supplementary Fig. 2a). To evaluate the IMC in PfPPP8-deficient parasites, we performed immunofluorescence against IMC-associated protein PfGAP45[16]. In PfPPP8-sufficient parasites, PfGAP45 surrounded each individual merozoite while in PfPPP8-deficient parasites, PfGAP45 surrounded multiple nuclei or formed anuclear compartments (Fig. 1d). The plasma membrane-localized PfMSP1[48] was similarly disorganized, and PfMORN1 did not appear as uniform rings or dots (Fig. 1e)[22,35]. The microneme protein PfAMA1 translocates to the plasma membrane around the time of PVM rupture. Thus, translocated PfAMA1 in E64-stalled schizonts indicates proper microneme secretion[45,46,49]. In E64-stalled PfPPP8-deficient parasites, PfAMA1 still colocalized with PfGAP45 following translocation (Fig. 1d), demonstrating that microneme discharge was unaffected. Transmission electron microscopy confirmed segmentation failure in PfPPP8-deficient schizonts (Fig. 1f and Supplementary Fig. 2b). E64-stalled PfPPP8-deficient schizonts generated few "megazoites"—massive merozoites containing most of the newly formed nuclei and apical organelles. In these abnormal parasites, apical organelle morphology was not disrupted (Fig. 1f and Supplementary Fig. 2b).

Because segmentation is paired with the final karyokinesis in *Plasmodium*[12], we wanted to see whether PfPPP8-knockdown modified the localization or distribution of the parasite centrosomes. There was, however, no significant difference in the appearance of the centrosome between PfPPP8-sufficient and -deficient parasites in slide-based or batch-stained IFAs (Supplementary Fig. 3a, b). Furthermore, the division of nuclei in PfPPP8-deficient parasites was comparable to PfPPP8-sufficient parasites (Supplementary Fig. 3a, b).

### PfPPP8 is essential for the stability of the basal complex

Immunofluorescence showed that in PfPPP8-deficient parasites, BC proteins PfMORN1 and PfBCP1 formed fragmented or nonuniform rings (Supplementary Fig. 4a)[15,22]. We tagged PfPPP8-interacting BC protein PfCINCH with the spaghetti monster-Myc (smMyc)[42] epitope tag in the PfPPP8-smV5$^{Tet}$ background, generating PfPPP8-smV5$^{Tet}$; PfCINCH-smMyc parasites. Visualizing PfCINCH-smMyc in PfPPP8-deficient parasites allowed us to clearly see BC rings with multiple breakage points, fragmented BC rings, and BC rings that varied significantly in size and shape within individual parasites, compared to the uniform BC rings of PfPPP8-sufficient parasites (Fig. 2a). When we tagged the BC protein PfBLEB in the PfPPP8-smV5$^{Tet}$ background[29] with HaloTag[50] (generating the PfPPP8-smV5$^{Tet}$; PfBLEB-HaloTag line) we also saw disorganized, fragmented PfBLEB-HaloTag rings in PfPPP8-deficient parasites, confirming PfPPP8 is essential for BC stability (Supplementary Fig. 4b).

To quantify differences in BC shape between PfPPP8-sufficent and -deficient parasites, and within individual PfPPP8-deficient parasites, we calculated the circularity of each visible BC ring in ten parasites per condition, using PfCINCH-smMyc as a BC marker. This measure (where proximity to 1 on a 0–1 scale indicates a more circular shape) allowed us to compare the jagged, irregular BCs of PfPPP8-deficient parasites to more uniform, rounder PfPPP8-sufficient BCs (Fig. 2b, c). The mean circularity of BC rings in PfPPP8-deficient parasites was significantly lower ($0.71 \pm 0.12$) than that of PfPPP8-sufficient parasites ($0.82 \pm 0.07$) ($p < 0.0001$), indicating defects in BC expansion or organization (Fig. 2b, c). Circularity measurements for individual parasites (Fig. 2b) also demonstrate that PfPPP8-deficient parasites have a greater range of circularity values (0.6954 vs 0.3815 for PfPPP8-sufficient), and an $F$-test confirms their variance is significantly greater ($p < 0.0001$). Thus, synchronous, uniform BC expansion is compromised in PfPPP8-deficient parasites.

Next, we utilized U-ExM[51] with a non-specific fluorescent protein stain (N-hydroxysuccinimide ester linked to AlexaFluor 405 [NHS-AF405][52], recently used to visualize the *P. falciparum* conoid[51,53]. In PfPPP8-smV5$^{Tet}$; PfCINCH-smMyc parasites, NHS-AF405 clearly delineates the BC, validated by anti-Myc and anti-V5 co-staining. In

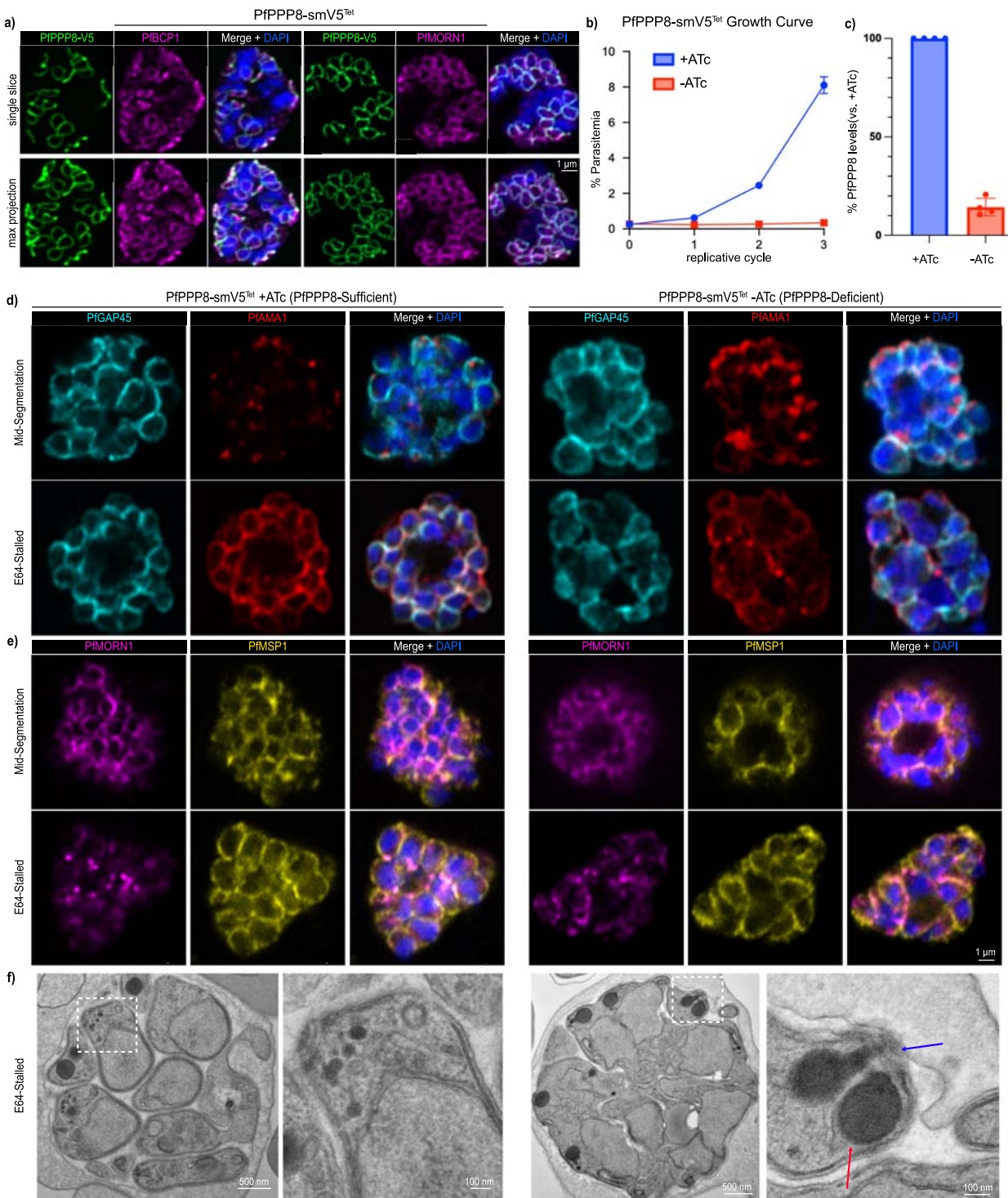

**Fig. 1 | PfPPP8 is a basal complex protein required for the formation of uniform merozoites during segmentation. a** Localization of PfPPP8 via immuno-fluorescence and colocalization with PfBCP1 (left images) and PfMORN1 (right images). Top row, individual slices; bottom row, maximum projections. **b** PfPPP8-smV5[Tet] replication curve with (PfPPP8-sufficient) and without (PfPPP8-deficient) ATc. Data shown as mean with 95% CI (*n* = 3). **c** Quantification of PfPPP8 knockdown by immunoblot. Data shown as mean ± SD (*n* = 4). **d** Visualization of PfPPP8-deficient phenotype with an inner membrane complex marker (PfGAP45) and apical organellar marker (PfAMA1) compared to PfPPP8-sufficient parasites mid-segmentation and stalled with E64 via immunofluorescence. **e** Visualization of PfPPP8-deficient phenotype with a parasite plasma membrane marker (PfMSP1) and a basal complex marker (PfMORN1) compared to PfPPP8-sufficient parasites mid-segmentation and stalled with E64 via immunofluorescence. **f** Visualization of PfPPP8-deficient phenotype via transmission electron microscopy of E64-stalled schizonts, demonstrating overall defective segmentation despite properly formed rhoptries (red arrow) and apical polar rings (blue arrow). White boxes on the image of the entire schizont (left image in each column) indicate the zoomed-in region depicted in the right image of each panel.

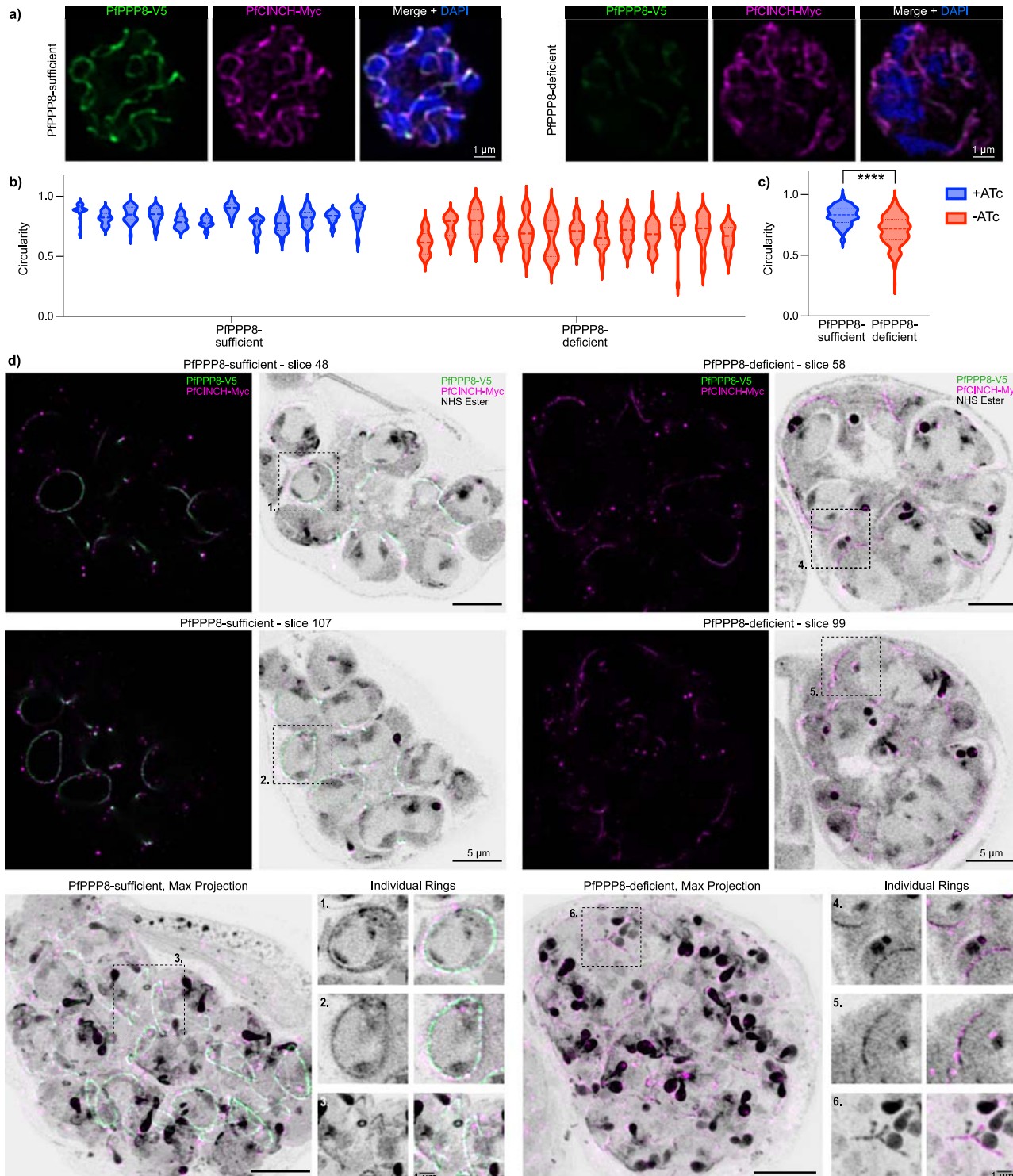

**Fig. 2 | The basal complex in PfPPP8-deficient parasites lacks circular uniformity and ring integrity. a** Comparison of basal complex phenotype (PfCINCH-smMyc) in PfPPP8-sufficient and -deficient parasites by IFA. **b** Quantification of basal complex circularity in individual PfPPP8-sufficient and -deficient parasites. Each violin represents a set of circularity values for a single parasite's basal complex rings ($n = 12$). **c** Quantification of basal complex circularity in grouped PfPPP8-sufficient and -deficient parasites; two-tailed unpaired Welch's $t$-test ($p < 0.0001$, $n = 230$ BC rings). Note that measurements of circularity do not utilize units and are always between 0 (least circular) and 1 (a perfect circle). **d** PfPPP8-sufficient and -deficient basal complex phenotype visualized by U-ExM. Top two rows show two individual slices from mid-segmentation PfPPP8-sufficient and PfPPP8-deficient schizonts; each slice is shown with only the epitope-specific fluorescent channels on the left and the fluorescent channels plus NHS-Ester on the right. Below, maximum projections of both parasites. Adjacent to each are images of single rings, isolated for emphasis and corresponding to the boxed areas of individual slices (boxes 1-3, PfPPP8-sufficient, boxes 4-6, PfPPP8-deficient).

U-ExM, PfPPP8-sufficient parasites possessed uniform and intact BC rings (Fig. 2d and Supplementary Movie 1).

Via U-ExM, the BC of PfPPP8-deficient parasites is extensively fragmented. Breaks in the BC are visible both by PfCINCH-smMyc and NHS-AF405—thus, these breaks separate the entire complex, rather than individual proteins (Fig. 2d and Supplementary Movie 2). Max projections show each BC "ring" has multiple breaks; unlike in regular IFA, no intact (lacking breaks) BC rings are visible by U-ExM. PfPPP8-deficient BC rings showed variably sized and extensively fragmented BC rings in three-dimensional structured illumination microscopy as well (Supplementary Fig. 5a). Immunoblotting of PfPPP8-smV5[Tet] ; PfCINCH-smMyc parasites showed PfPPP8-deficient parasites had PfMORN1/ PfCINCH levels comparable to PfPPP8-sufficient parasites, suggesting that increased BC fragmentation is not due to lack of expression of other BC proteins (Supplementary Fig. 5b, c).

## PfPPP8 is not required for initial BC biogenesis
At the achievable level of knockdown, young, small BC rings of PfPPP8-deficient early segmentation stage schizonts appeared uniform but never maximally expanded or contracting rings (Fig. 3a–d). We hypothesized that BC fragmentation in PfPPP8-deficient parasites could be time-dependent, based on the stage of segmentation. To evaluate this, we synchronized PfPPP8-smV5[Tet]; PfCINCH-smMyc parasites within a 2-h window. The BC was visualized via immunofluorescence every 2 h during segmentation (40–48 hpi). For ≥6 parasites per time point and condition, the diameter of every ring (intact or with a ≤0.4 μm break) was measured, totaling >100 each (Representative parasites: Fig. 3a). During initial BC formation (40 hpi), there was no significant difference between PfPPP8-sufficient and -deficient parasites in mean BC diameter (BCD) or BCD variance.

From 42 to 48 hpi, BCD variance increased in PfPPP8-deficient vs PfPPP8-sufficient parasites, indicating failure to uniformly expand in the latter (Fig. 3b). BCD variance in PfPPP8-deficient parasites became greater at 42 hpi ($p = 0.0199$), significantly greater at 44, 46, and 48 hpi ($p = <0.0001$) (Fig. 3b.). At 46 hpi, when PfPPP8-sufficient parasites' BC reached a maximum diameter (mean = $0.96 \pm 0.08$; range = 0.36 μm), PfPPP8-deficient parasites' mean BCD was $0.73 \pm 0.32$ μm, with a range of 1.7 μm (Fig. 3b). In PfPPP8-deficient parasites, mean BCD only decreased from $0.73 \pm 0.32$ to $0.60 \pm 0.31$ μm between 46 and 48 hpi, versus decreasing from $0.96 \pm 0.08$ μm to $0.13 \pm 0.11$ μm in PfPPP8-sufficient schizonts, illustrating contraction failure (Fig. 3b). Individual intra-parasite BCD variances are compared in Supplementary Fig. 5d.

We then optimized a protocol for long-term live cell fluorescent imaging in *P. falciparum* to visualize BC formation, expansion, and contraction[50,54]. Using PfPPP8-smV5[Tet]; PfBLEB-HaloTag, we took 3-dimensional images every 20 min and measured the diameter of each BC ring in pairs of parasites matched by initial mean BCD (Fig. 3c, d and Supplementary Movie 3 (PfPPP8-sufficient) and Supplementary Movie 4 (PfPPP8-deficient)). An additional pair is shown in Supplementary Fig. 5e, f and Supplementary Movie 5 (PfPPP8-sufficient), and Supplementary Movie 6 (PfPPP8-deficient).

BCD variance in the PfPPP8-deficient parasite started to increase at TP 1:20, 1 h and 20 min after TP 0:00, when the PfPPP8-sufficient parasite's mean BCD was $0.79 \pm 0.07$ μm. From time point (TP) 2:00 to TP 2:20, the BCD standard deviation in the PfPPP8-deficient parasite increased from 0.086 to 0.18 μm; its BC rings expanded nonuniformly (Fig. 3c, d). In the PfPPP8-sufficient parasite, the BCD standard deviation increased from 0.055 to 0.073. At TP 2:20, the PfPPP8-sufficient parasite's BCD was greatest ($1.07 \pm 0.07$ μm) and their difference in BCD variance became statistically significant ($p = 0.0005$). The PfPPP8-deficient parasite had a 1.23 μm range in BCD at TP 2:20, almost 7-fold higher than the PfPPP8-sufficient parasite's (range: 0.177 μm). As the PfPPP8-sufficient parasite's BC contracted, the PfPPP8-deficient parasite's BCD variance increased ($p < 0.0001$ from TP 2:40 onward) (Fig. 3c, d). BCD variance in the PfPPP8-deficient parasite remained

significantly larger than in the PfPPP8-sufficient parasite ($p < 0.0001$) (Fig. 3c, d). These data corroborate fixed immunofluorescence timeline data; PfPPP8 is required to maintain BC integrity and uniform expansion.

## PfPPP8 is a founding pseudophosphatase within the PPP family
Automated bioinformatic analysis identified a PPP-type serine-threonine protein phosphatase domain in PfPPP8[7], but manual investigation of its sequence revealed that catalytic residues of one core motif (GDXXDRG) are mutated (GNLINRG) (Supplementary Fig. 6a, b)[41,55–57]. These aspartic acids coordinate metal ions[58–60], and their alteration disrupts phosphatase activity[61,62]. PfPPP8 is thus a pseudophosphatase —a protein with a phosphatase domain but lacking enzymatic activity— and the first pseudophosphatase identified in the PPP family[36].

PfPPP8 homologs in *Cryptosporidia parvum*, *Babesia bovis*, and *Theileria parva* have been identified in silico[63,64]. We used these homologs as seeds for protein-BLAST searches of PiroPlasmDb, EuPathDb, PlasmoDB, and ToxoDB, and investigated extant literature[31,63,65,66]. Homologs were identified across and beyond Apicomplexa (Supplementary Table 1). All contained bioinformatically predicted serine-threonine protein phosphatase domains[56,57] (Supplementary Fig. 6b). Most homologs possessed mutations in critical catalytic motifs, but even those with seemingly intact motifs like TgBCC5 had different numbers of amino acids between them than PPP-type phosphatase *S. cerevisiae* PPT1 (Supplementary Fig. 6b and Supplementary Note 1)[7,31,65,67,68]. PfPPP8 orthologs were also found in free-living protists *Vitrella brassicaformis* and *Chromera velia* which share a common ancestor with Apicomplexans—thus, despite significant differences in lifestyle and a dearth of knowledge about the cell biology of colpodellids, mechanisms of division are likely to be similar even if, as with *T. gondii* and BCC5, PfPPP8's homolog does not play an identical role (Supplementary Table 1)[65,69,70].

Some pseudophosphatases can attain phosphatase activity when catalytic residues are restored[37,71]. We recombinantly expressed phosphatase domains of PfPPP8 WT, PfPPP8[N1311,I314D], CpPPP8, CpPPP8[G406D,445R] and *S. cerevisiae* PPT1 with C-terminal 6xHis-tags (Supplementary Fig. 6c) and, separately, with N-terminal maltose-binding protein (MBP)-tags, which generated a larger amount of more pure recombinant protein (Supplementary Fig. 6d). Both 6xHis-ScPPT1 and MBP-ScPPT1 dephosphorylated p-nitrophenyl phosphate but no recombinant Apicomplexan phosphatase domains could, regardless of catalytic residue identity (Supplementary Fig. 6e, f)[72]. Considering the differences in the number of amino acids present between conserved motifs between PfPPP8 (and other Apicomplexan homologs) and ScPPT1, this family of pseudophosphatases could potentially lack activity because they differ structurally from active Ser/Thr phosphatases in ways that preclude activation by sequence changes alone (Supplementary Note 1).

## PfPPP8 interacts with other members of the basal complex including PfMyoJ and PF3D7_0214700
Because few *Plasmodium* BC proteins have been validated, we wanted to identify more components of this multi-protein molecular machine[31,32]. Tightly synchronized PfPPP8-smV5[Tet]; PfCINCH-smMyc parasites (and the parental strains) were collected at mid-segmentation and subjected to co-immunoprecipitation. Following two biological replicates, we identified and compared consensus proteins for PfPPP8-V5 and PfCINCH-Myc (Fig. 4a, b).

Almost 50 proteins were shared between the consensus lists (Supplementary Table 2). From the proteins enriched >50-fold compared to the control, we selected two for validation (Fig. 4b and Supplementary Table 2): PF3D7_1229800/PfMyoJ, because its homolog, TgMyoJ, is in the *T. gondii* BC[31,34,73] and PF3D7_0214700 because it was uncharacterized and unique to *Plasmodium* (and *Hepatocystis*)[65]. While examining PfMyoJ could identify conserved segmentation

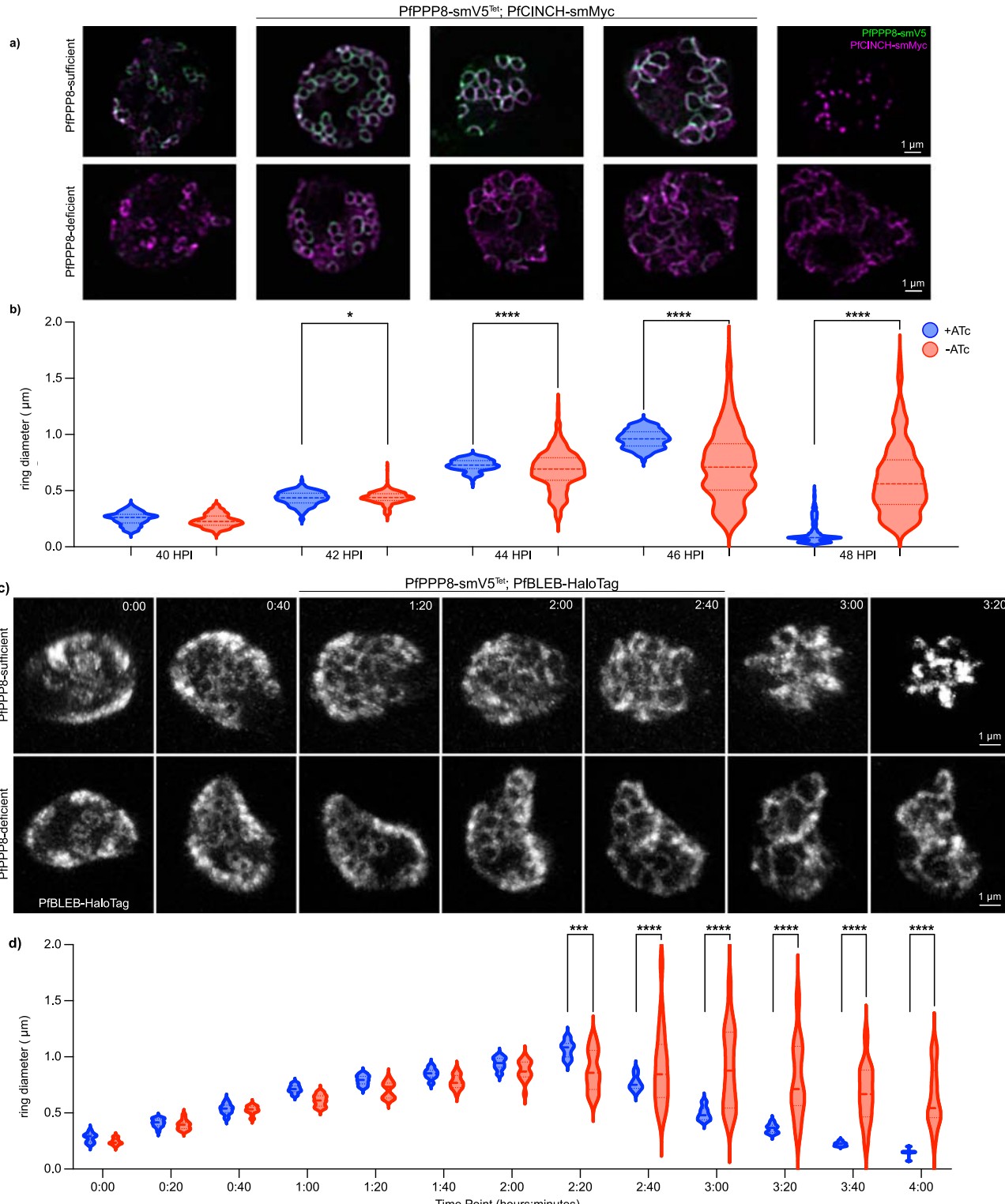

**Fig. 3 | The basal complex in PfPPP8-deficient parasites initially resembles that of PfPPP8-sufficient parasites, but does not expand uniformly, accumulates breaks, and collapses. a** Comparison of basal complex phenotype (PfCINCH-smMyc) in PfPPP8-sufficient and -deficient parasites by IFA throughout segmentation. **b** Quantification of basal complex "ring" diameter in grouped PfPPP8-sufficient and -deficient parasites at 2-h intervals during segmentation; one-sided *F*-test for equality of variances (42 hpi: *p* = .02; 44–48 hpi: *p* < 0.0001, *n* > 100 BC

rings). **c** Selected time points (time represented as h:min on each image) showing basal complex phenotype (PfBLEB-HaloTag) in the same parasite (PfPPP8-sufficent and -deficient) over time (see Supplementary Movies 3 and 4). **d** Quantification of basal complex "ring" diameter in (**c**) parasites over the course of imaging; one-sided *F*-test for equality of variances (2:20: *p* = 0.0005; 2:40 onward: *p* < 0.0001, *n* = 10–25 BC rings). Time point = time after initiation of imaging.

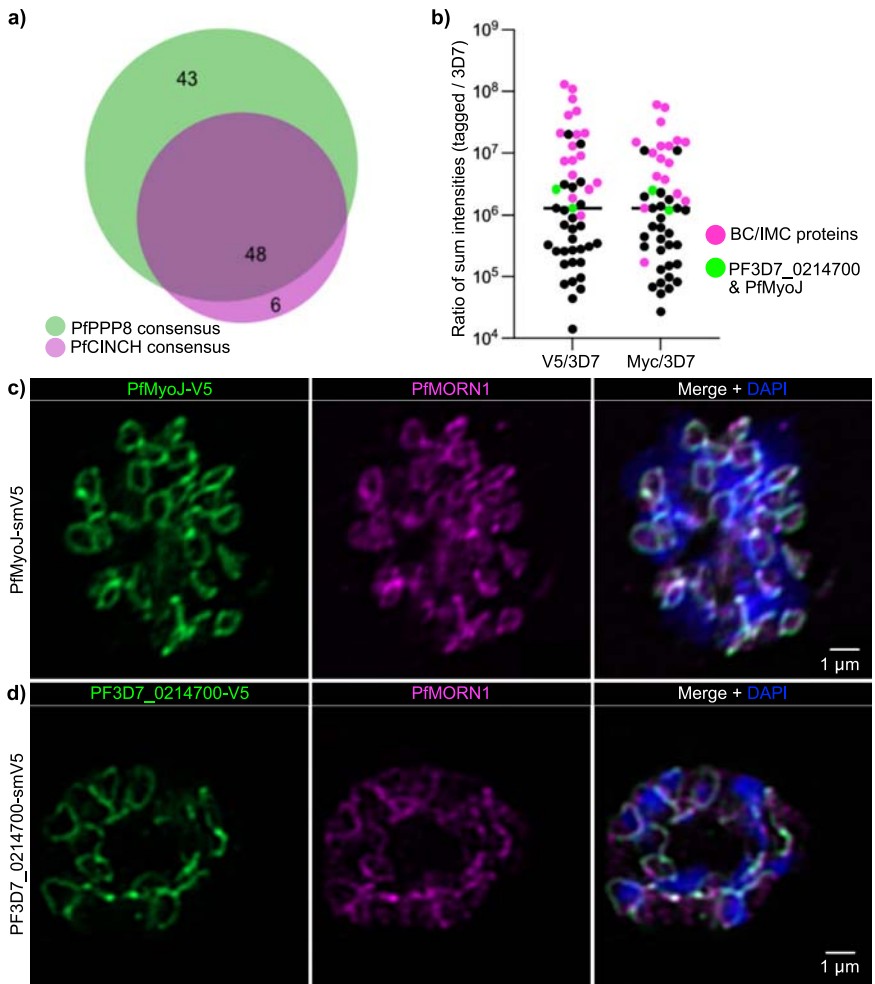

**Fig. 4 | PfMyoJ and PF3D7_0214700, identified via immunoprecipitation against PfCINCH-smMyc and PfPPP8-smV5, localize to the basal complex.**
a Proportional Venn diagram of the consensus (present in both replicates) hits for both PfPPP8-smV5 and PfCINCH-smMyc. b Dot plot depicting the ratio of sum intensities of the 48 consensus proteins from (a) for each sample over its 3D7 control for replicate one. Magenta dots = experimentally confirmed or bioinformatically predicted[29,83] basal complex and inner membrane complex localized proteins. Green dots = PF3D7_0214700 and PfMyoJ. Black dots = all other (i.e., non-BC/IMC localized proteins) consensus proteins. c Visualization of PfMyoJ-smV5 localizing to the basal complex (marked by PfMORN1) via IFA. d Visualization of PF3D7_0214700-smV5 localizing to the basal complex (marked by PfMORN1) via IFA.

mechanisms between Apicomplexans and was likely to be localized to the BC, we were interested in PF3D7_0214700 because of its limitation to *Plasmodium/Hepatocystis* spp. As a unique protein, characterization of PF3D7_0214700 could elucidate mechanisms of segmentation, aspects of BC composition, and BC functions specific to the sub-Apicomplexan *Plasmodiidae* family[10]. Furthermore, as one of the smallest yet-uncharacterized proteins we identified (35kD), PF3D7_0214700 lends itself readily to future in-vitro experimentation (Supplementary Table 2). We thus generated C-terminally tagged PfMyoJ-smV5 and PF3D7_0214700-smV5 strains; both colocalized with PfMORN1, validating their identification as BC proteins (Fig. 4c, d).

### PfMyoJ and PF3D7_0214700 are recruited to the basal ring when maximally expanded

Neither PfMyoJ nor Pf3D7_0214700 were detectable by immunofluorescence in early segmentation schizonts (~40 hpi) when BC rings are small and appear attached (Supplementary Fig. 7a, d). This suggested that PfMyoJ and PF3D7_0214700's BC localizations are restricted to later segmentation. In *T. gondii*, TgMyoJ and TgCen2 are recruited to the BC after TgMORN1[10,25,31,34]. We imaged tightly synchronized parasites every 2 h from 40 to 48 hpi and treated with Protein Kinase G inhibitor, Compound 1, to prevent egress[46] (Fig. 5a, b). Each time point we took ≥10 immunofluorescence images, measured

ten BC rings' diameter per schizont, and counted the number of PfMyoJ-smV5 or PF3D7_0214700-smV5-positive parasites from 100 PfMORN1-positive schizonts (Fig. 5c, d and Supplementary Fig. 7c, f). At 40 hpi, no PfMORN1-positive schizonts were PfMyoJ-smV5-positive; by 48 hpi 84 ± 5.3% of PfMORN1-positive schizonts were PfMyoJ-smV5-positive (Fig. 5c). Similarly, at 40 hpi no PfMORN1-positive schizonts were PF3D7_0214700-smV5 positive; by 48 hpi, 78.8 ± 5.5% of PfMORN1-positive schizonts were PF3D7_0214700-smV5-positive (Fig. 5d). Since synchronization is imperfect, smV5-negative "48-hpi" parasites were likely earlier schizonts.

PfMyoJ and PF3D7_0214700 are therefore recruited to the BC around 46 hpi, when the BC is maximally expanded (PfMyoJ-smV5 mean BCD 0.92 ± 0.08 µm; PF3D7_0214700-smV5 mean BCD 0.93 ± 0.15 µm) and are present during contraction (48 hpi). PfMyoJ-smV5 protein levels increased over tenfold from 42 to 48 hpi, compared to PfMORN1 (Supplementary Fig. 7b). PF3D7_0214700 protein levels increased threefold from 42 to 48 hpi (Supplementary Fig. 7e). Thus, PF3D7_0214700 and PfMyoJ are recruited to the BC at mid-segmentation and actively synthesized during schizogony.

Since both new proteins are recruited to the maximally expanded BC, we wanted to determine if they could be recruited to the morphologically defective BC of PfPPP8-deficient schizonts and generated the lines: PfPPP8-smV5^Tet; PfMyoJ-smMyc and PfPPP8-smV5^Tet;

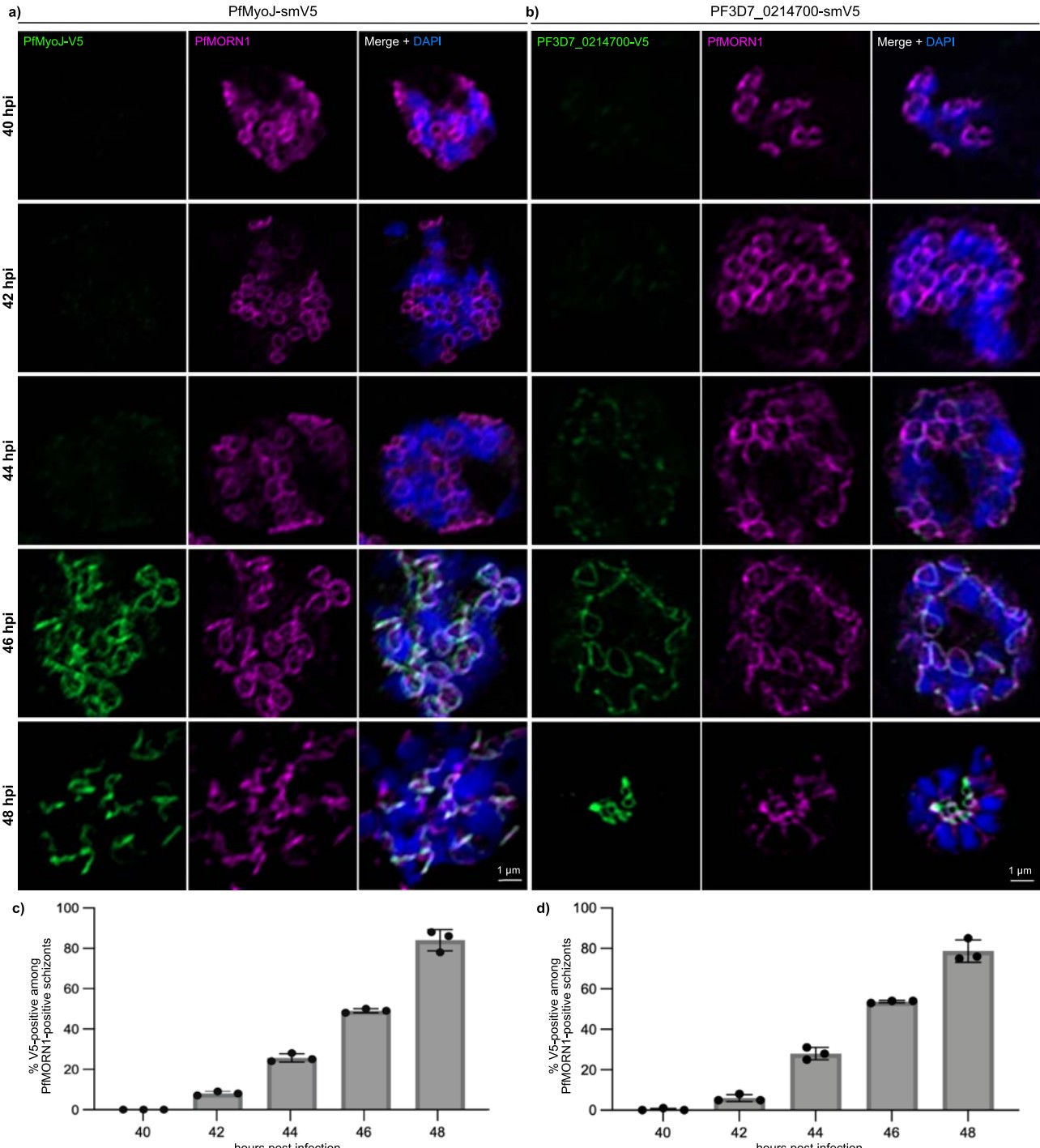

**Fig. 5 | PfMyoJ and PF3D7_0214700 are recruited to the basal complex at the midpoint of segmentation when the basal complex diameter is greatest.**
**a** Representative images of PfMyoJ-smV5 parasites at 2-h time points during segmentation, colocalizing with PfMORN1 at 46 and 48 hpi only. **b** Representative images of PF3D7_0214700-smV5 parasites at 2-h time points during segmentation, colocalizing with PfMORN1 at 46 and 48 hpi only. **c** Quantification of PfMORN1-positive schizonts also positive for PfMyoJ-smV5 at each time point. **d** Quantification of PfMORN1-positive schizonts also positive for PF3D7_0214700-smV5 at each time point. *N* = 100 parasites for 3 biologically independent experiments. Data for (**c**) and (**d**) presented as mean values ± SD; *n* = 100 parasites for 3 biologically independent experiments.

PF3D7_0214700-smMyc. PfMyoJ-smMyc colocalized with PfPPP8-smV5 and PfMORN1 in both PfPPP8-sufficient and -deficient parasites, the latter demonstrating that PfMyoJ was still recruited to the BC (Supplementary Fig. 8a). PfMyoJ remained at the IMC's leading edge in late segmentation PfPPP8-deficient schizonts, localizing to the disorganized, fragmented BC (Supplementary Fig. 8b). PF3D7_0214700 was also still recruited to the fragmented, nonuniform BC of PfPPP8-

deficient parasites, colocalizing with remaining PfPPP8-smV5 and localizing to the edge of the IMC (Supplementary Fig. 8c).

## PfPPP8 is degraded at the end of segmentation and is not present in merozoites
PfPPP8 was never visualized at the basal end of merozoites once segmentation was complete, unlike PfCINCH-smMyc or PfBLEB-HaloTag

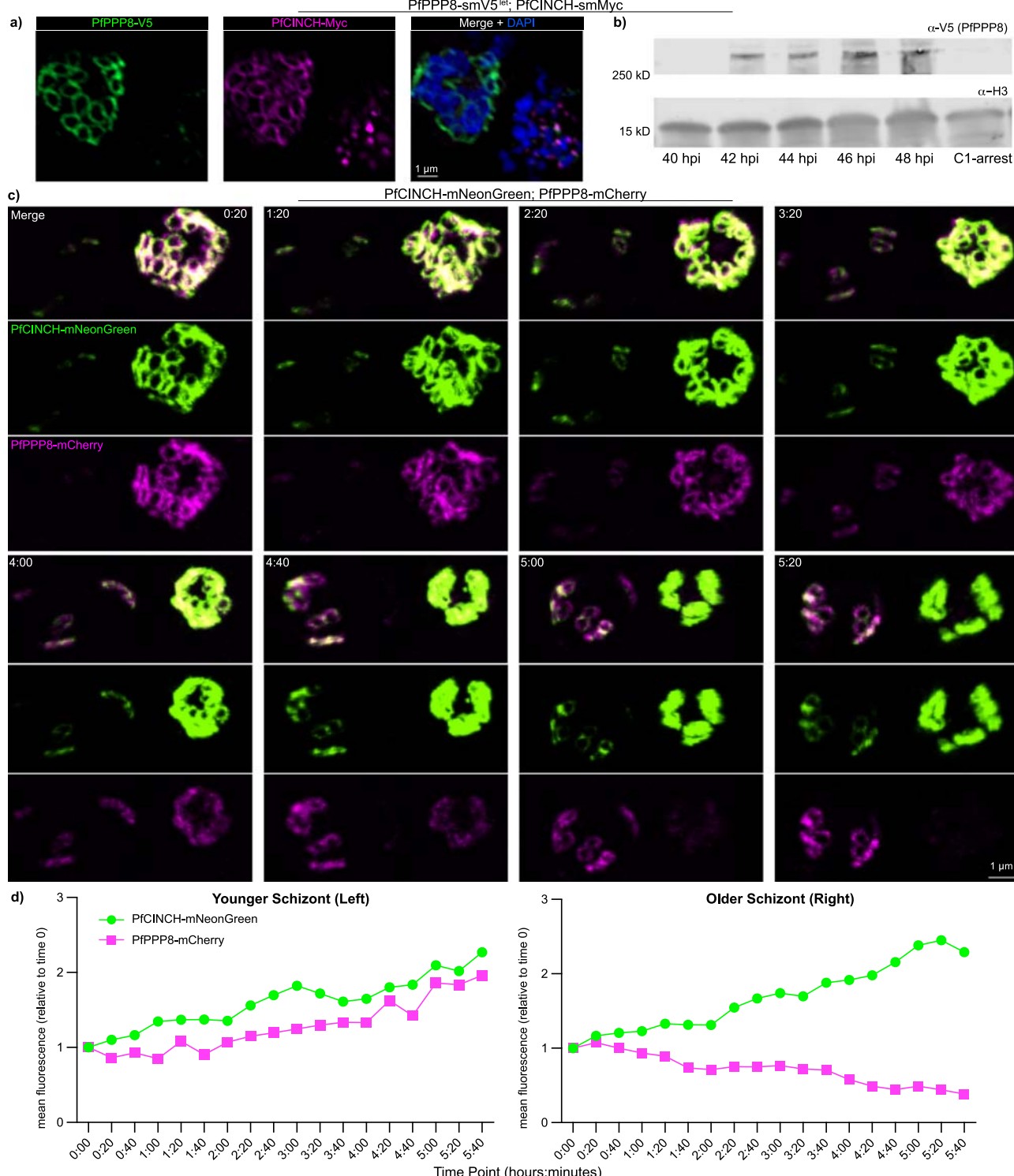

**Fig. 6 | PfPPP8 is removed from the basal complex as segmentation concludes and is fully degraded before egress. a** Comparison of mid-segmentation (top) and post-segmentation (bottom) PfPPP8-smV5^Tet; PfCINCH-smMyc parasites by IFA in a mixed-schizont population. **b** Immunoblot against PfPPP8-smV5^Tet parasites comparing the amount of PfPPP8-smV5 present in the parasite population at different time points throughout segmentation. **c** Selected time points from Supplementary

Movie 7 comparing PfPPP8-mCherry and PfCINCH-mNeonGreen expression in parasites that begin imaging in early (left) and mid (right) segmentation (time represented as h:min on each image). **d** Quantification of relative fluorescence, compared to fluorescence at the onset of imaging, for the individual earlier (left) and later (right) parasites over the course of imaging. Time point = time after initiation of imaging.

(Fig. 6a and Supplementary Fig. 9a, b). Tight synchronization of PfPPP8-smV5^Tet; PfCINCH-smMyc parasites confirmed PfPPP8 was absent at 48 hpi, though at 40 hpi, only dual-positive (PfPPP8-smV5; PfCINCH-smMyc) parasites were seen (Supplementary Fig. 9b).

Immunoblotting of Compound 1-stalled schizonts showed PfPPP8, unlike PfCINCH and PfMORN1, is absent post-contraction, thus it is likely actively degraded (Fig. 6b and Supplementary Fig. 9c). In a U-ExM time course of PfPPP8-smV5^Tet; PfCINCH-smMyc parasites,

PfPPP8-smV5 was depleted during BC contraction, with dramatically reduced signal well before egress (Supplementary Fig. 10). PfPPP8 was undetectable in a schizont with merozoites still connected to the residual body; therefore PfPPP8-degradation precedes egress (Supplementary Fig. 10e).

To corroborate this, we generated PfCINCH-mNeonGreen; PfPPP8-mCherry parasites and imaged a mixed-schizont population over 5 h. PfPPP8 and PfCINCH's temporal expressions were compared by examining adjacent schizonts that began imaging at different stages of segmentation—the left early, the right later (Fig. 6c and Supplementary Movie 7). We determined the mean fluorescence intensity (MFI) for each fluorophore at each time point and plotted the relative change in each protein's MFI compared to TP0. PfCINCH-mNeonGreen's MFI increased during contraction, inversely correlated with ring diameter, as did the MFI of late segmentation PfPPP8-smV5[Tet]; PfBLEB-HaloTag parasites (Fig. 6c and Supplementary Fig. 9d, f). In contrast, PfPPP8-mCherry's MFI decreased significantly during contraction, consistently across replicates (Fig. 6c, d, Supplementary Fig. 11c, d and Supplementary Movies 7 and 8).

Juxtaposition with an earlier-hpi schizont where PfPPP8-mCherry remains bright renders photobleaching less likely to cause MFI decrease in the older, contracting schizont (Fig. 6c, d and Supplementary Movie 7)[74–76]. In younger schizonts, PfPPP8-mCherry's MFI increases during imaging as the BC expands and PfPP8-mCherry and PfCINCH-mNeonGreen assemble and expand the BC (Fig. 6c, d). Early segmentation parasites imaged over 9 h demonstrate consistent MFIs for PfPPP8-mCherry and PfCINCH-mNeonGreen (Supplementary Fig. 11a, b and Supplementary Movie 9).

Averaging MFI across multiple replicates demonstrated a steady decrease during late segmentation for PfPPP8-mCherry—regression analysis returned a negative slope (Supplementary Fig. 11c). PfCINCH-mNeonGreen's MFI consistently increased in late segmentation—regression analysis returned a positive slope (Supplementary Fig. 11c). Averaging multiple PfPPP8-smV5[Tet]; PfBLEB-HaloTag parasites showed increasing MFI for late segmentation PfBLEB-HaloTag parasites as well (Supplementary Fig. 9d–f). To confirm our findings were not impacted by our choice of fluorophore for PfPPP8, we constructed another line: PfCINCH-mScarlet; PfPPP8-mNeonGreen. Again, in individual and averaged late segmentation schizonts, the MFI of PfCINCH-mScarlet increases while PfPPP8-mNeonGreen's MFI decreases (Supplementary Fig. 12a–d and Supplementary Movie 10). In a younger schizont, with newly formed BCs at TP0, PfCINCH-mScarlet and PfPPP8-mNeonGreen's MFI both increase. When a contraction begins (TP 7:20), the MFI of PfPPP8-mNeonGreen declines to below even the TP0 MFI. (Supplementary Fig. 12e and Supplementary Movie 11). These data corroborate our U-ExM results, showing PfPPP8 begins to be degraded during contraction and is undetectable before egress.

## Discussion

We determined that PfPPP8, a novel BC protein in *P. falciparum*, is essential for maintaining BC integrity and uniformity. With U-ExM we confirmed that in PfPPP8-deficient schizonts, BC fragmentation was extensive throughout the entire BC; U-ExM revealed breakages too small to distinguish via conventional immunofluorescence.

There is increased intra-parasite BCD variance in PfPPP8-deficient parasites after 44 hpi. Their BCs do not uniformly expand; evident when the BCD reaches 0.7–0.85 microns. Long-term, live cell microscopy of individual parasites showed this significant increase in intra-parasite BCD variance by mid-segmentation, with breakages accumulating throughout segmentation. PfPPP8-sufficient parasites maintained low intra-parasite BCD variance. While a more complete PfPPP8 knockout could potentially abrogate initial complex formation, PfPPP8 is ~85% depleted upon knockdown, and many attempts to generate an inducible knockout have been unsuccessful.

PfPPP8 was also identified as a pseudophosphatase—the first experimentally characterized PPP-type serine-threonine pseudophosphatase and one of few *Plasmodium* pseudophosphatases[77]. Neither PfPPP8 nor its *Cryptosporidium* homolog were, or became through restoration of catalytic residues, enzymatically active. However, even PfPPP8's homologs without obvious mutations in catalytic motifs (TgBCC5 and VbPPP8) had more amino acids between motifs than active PPP-type phosphatase ScPPT1. This Apicomplexan pseudophosphatase family could be significantly diverged from active phosphatases, incapable of being easily rendered active as a result of larger structural changes to the phosphatase domain(s)[36].

Whether most of PfPPP8's homologs function similarly in asexual division remains unknown. However, TgBCC5 was proven nonessential to *Toxoplasma* endodyogeny – TgBCC5-KO parasites are only deficient in intravacuolar communication between tachyzoites, a significant divergence from PfPPP8's importance to schizogony[31,70]. Furthermore, TgBCC5 is present in mature tachyzoites' BC, whereas PfPPP8 is degraded before schizogony ends[70]. TgBCC4-deficient parasites do resemble PfPPP8-deficient schizonts, surprisingly, since TgBCC4 is structurally distinct from PfPPP8 and lacks *Plasmodium* homologs. TgBCC4 also leaves the *T. gondii* BC before TgCEN2/TgMyoJ recruitment, unlike PfPPP8 which departs before egress, but after PfMyoJ recruitment[31]. Likely, the coordinated construction of dozens of BCs in schizogony requires novel and/or specially adapted protein(s) for maintaining both stability and synchrony.

In *T. gondii*, recent studies have suggested a model where the BC expands and stretches like a rubber band until it reaches its maximal diameter at the midpoint of segmentation[26,31]. PfPPP8-deficient schizonts' increased BCD mean and variance suggests they exceed this "maximal stretch", resulting in BC fragmentation—the rubber band "snaps". PfPPP8 could thus act to signal this "maximal stretch" point. Aside from limiting BC "stretch" to a consistent maximal diameter, PfPPP8 is essential for uniform BC expansion. In *Plasmodium* schizogony, BCs expand like *multiple* rubber bands simultaneously stretched to the same degree. This requires communication between developing merozoites, linking PfPPP8 functionally to TgBCC5 and intravacuolar communication between tachyzoites[70]. Since *T. gondii* cytokinesis happens in cycles rather than simultaneously, this communication is less necessary and TgBCC5 less essential than PfPPP8, although they may be acting in similar processes.

The presence of PfPPP8 homologs across Apicomplexa indicates an ancient origin and a broad importance to multiple division modalities. Endopolygeny in *Cystoisospora*, for example, requires the coordination of dozens of BCs, like schizogony[10]. PfPPP8 homologs in photosynthetic, free-living protists suggest that the BC as a cytokinetic structure evolutionarily predates parasitism, and a PfPPP8-like protein could be essential to the "original" BC's formation.

PfPPP8's structure allows us to speculate about potential mechanisms. Although two Asp to Asn mutations render PfPPP8 catalytically inactive, its PPP-type phosphatase domain is otherwise intact. The negatively charged asparagine residues mutated in PfPPP8 coordinate calcium ions during dephosphorylation in active PPP-type phosphatases[58]. Since the positively charged amino acids in other motifs have not been modified, PfPPP8 could theoretically bind phosphorylated protein(s) in early segmentation, releasing it upon depletion as part of the tightly regulated cascade of events leading to egress. Indeed, *C. elegans* pseudophosphatases EGG-3 and EGG-4/5 bind and sequester kinase MBK-2, controlling its activity during the oocyte to zygote transition[78,79]. MBK-2 is released only when EGG-3 is degraded[80]. PfPPP8 could similarly function to temporarily "trap" phosphorylated substrate(s) until its degradation during late segmentation.

Dual co-immunoprecipitation identified PfMyoJ and PF3D7_0214700 as BC proteins. PfMyoJ's localization to the BC and late

recruitment mimics TgMyoJ. While PfMyoJ was predicted to be non-essential and demonstrated to be nonessential for asexual blood stages in *P. berghei*, TgMyoJ-KO tachyzoites have incomplete BC contraction. PfMyoJ may similarly contribute to, despite not driving, BC contraction[34,40,81]. PF3D7_0214700 has a similar temporal localization, though it lacks identifiable domains and *Toxoplasma* homologs[7,40,56]. It is predicted to be essential and could be a useful phenotypic comparand for PfPPP8 and PfCINCH. Interestingly, both proteins still localize to the BC in PfPPP8-deficient schizonts, indicating that synchronous expansion is not necessary for later recruitment—considering BC breakage points appear frequently in 46 hpi schizonts, not even an intact BC is necessary. This indicates also that PfPPP8 does not comprise an essential part of the signal for PfMyoJ/Pf3D7_0214700 recruitment. Though TgBCC4's absence also induces fragmentation of the *Toxoplasma* BC, whether this fragmentation hampers TgMyoJ and TgCen2 recruitment remains to be determined.

PfPPP8's temporal localization contrasts with PfMyoJ and PF3D7_0214700's. U-ExM and long-term live cell imaging determined that PfPPP8's departure from the BC begins during contraction. PfPPP8's decrease in MFI contrasts with the concurrent MFI increase of PfBLEB-HaloTag and PfCINCH-mNeonGreen. Without protein depletion, this inverse relationship between BCD and MFI is expected as the same amount of protein is packed into a now-smaller area. When relative MFIs of PfCINCH-mNeonGreen, PfBLEB-HaloTag, and PfPPP8-mCherry in contracting BCs are averaged, linear regression returns a positive slope for PfCINCH-mNeonGreen and PfBLEB-HaloTag, and a negative slope for PfPPP8-mCherry. The pattern of PfCINCH's MFI increasing and PfPPP8's decreasing during contraction holds true when the fluorophores are switched; PfPPP8-mNeonGreen's mean fluorescence decreases during contraction just as PfPPP8-mCherry's does, strongly corroborating previous evidence.

Identification of unique, disparate temporal localizations for all 3 proteins establishes the *P. falciparum* BC as a dynamic structure. Proteins including PfMORN1, PfBCP1, and PfCINCH are present in the BC for segmentation's duration. Proteins including PF3D7_0214700 and PfMyoJ are recruited in mid-segmentation, and proteins including PfPPP8 are present in the BC in early segmentation but depart before contraction is complete, a model depicted in Supplementary Fig. 13.

In conclusion, we identified PfPPP8 as an essential member of the *P. falciparum* BC, required for maintaining uniformity and integrity. We also identified PfPPP8 as the founding member of a PPP-type pseudophosphatase family with homologs throughout and beyond Apicomplexa. We identified PfMyoJ and PF3D7_0214700 as members of the BC and, along with PfPPP8, identified their unique temporal localizations. This establishes a strong foundation for further *P. falciparum* BC studies determining the order of assembly and disassembly of proteins within the BC.

## Methods
### Plasmid construction
**PfPPP8-smV5^Tet (pRR206).** The 3' homology region (HR) and 5'HR were PCR amplified from 3D7 genomic DNA (gDNA) with oJDD4692/4691 and oJDD4688/4689, respectively. A codon-altered region was appended onto the 5'HR by nested PCR with oJDD4688/4690. The two fragments were spliced by overlap-extension PCR (PCR SOEing) with oJDD4692/4690 and cloned as a NotI/NcoI fragment into pRR92[15] (which contains the smV5 epitope tag, ten copies of the tet-aptamer, and an expression cassette for human dihydrofolate reductase (hDHFR) and the TetR-DOZI fusion protein).

**PfCINCH-smMyc (pAM27).** SmMyc was PCR amplified with oJDD4293/4925 and cloned as a NcoI/XmaI fragment into pRR206. The hDHFR cassette in this intermediate plasmid was removed by excision with AflII/AvrII and replaced with a blasticidin-S-deaminase expression cassette. The PfCINCH-targeting regions from pRR92 were excised

with NotI/NcoI and cloned into the second intermediate plasmid using the same sites.

**PfBLEB-HaloTag (pAM99).** The PfBLEB-targeting regions from pJDD342[29] were excised with NotI/NcoI and cloned into pAM27. Then, HaloTag was excised from pRR223 (PfCINCH-HaloTag) with NcoI/KpnI and ligated into the intermediate plasmid using the same sites.

**PfCINCH-mNeonGreen (pRR208).** mNeonGreen was PCR amplified with oJDD4616/4617 and cloned as a XhoI/BsiWI fragment into pRR198, cut with XhoI and Acc65I.

**PfPPP8-mCherry (pAM110).** The 5' half of mCherry was amplified with oJDD6503/6504. The 3' half of mCherry was amplified with oJDD6505/6506. The two fragments were spliced by PCR SOEing with oJDD6503/6506, removing an internal NcoI site, then cloned as a NcoI/KpnI fragment into pAM27. The PfPPP8-targeting regions from pRR206 were then excised with NotI/NcoI and cloned into the intermediate plasmid using the same sites.

**PfMyoJ-smV5^Tet (pAM145).** The 3' HR and 5'HR of PfMyoJ were PCR amplified from 3D7 gDNA with oJDD6358/6359 and oJDD6360/6361, respectively. A codon-altered region consisting of the last 70 amino acids was generated by IDT DNA technologies, amplified using oJDD6362/6363, and sewed onto the 5'HR by nested PCR with oJDD6360/6363. The joined fragments were spliced by PCR SOEing with oJDD4692/4690 and cloned as a NotI/NcoI fragment into pRR92[15].

**PF3D7_0214700-smV5^Tet (pRR205).** The 3'HR and 5'HR of PF3D7_0214700 were PCR amplified from 3D7 gDNA with oJDD4699/4700 and oJDD4701/4702, respectively. A codon-altered region was appended onto the 5'HR by nested PCR with oJDD4701/4703. The two fragments were spliced by overlap-extension PCR SOEing with oJDD4699/4703 and cloned as a NotI/NcoI fragment into pRR92[15].

**PfMyoJ-smMyc (pAM159).** The homologous regions (3' HR and 5' HR) of PfMyoJ were PCR amplified from pAM145 with oJDD6358/6361 and cloned as a NotI/NcoI fragment into pAM27, displacing the previously present PfCINCH homologous regions.

**Pf3D7_0214700-smMyc (pAM160).** The homologous regions (3' HR and 5' HR) of Pf3D7_0214700 were PCR amplified from pRR205 with oJDD4699/4702 and cloned as a NotI/NcoI fragment into pAM27, displacing the previously present PfCINCH homologous regions.

**PfPPP8-mNeonGreen (pAM166).** The 5' half of mNeonGreen was amplified with oJDD7286/7287. The 3' half of mNeonGreen was amplified with oJDD7288/7289. The two fragments were spliced by PCR SOEing with oJDD7286/7289, removing an internal KpnI site, then cloned as a NcoI/KpnI fragment into pAM110.

**PfCINCH-mScarlet (pRR209).** mScarlet was amplified with oJDD4614/4615 and cloned as an XhoI/KpnI fragment into pRR92.

**PfPPP8 [AAs 1115-1641]-6xHis (pAM101).** A codon-altered portion of PfPPP8 containing the phosphatase domain was generated by IDT DNA Technologies (AAs 1211-1569[56,57]) and amplified with oJDD6246/6247. The pET28b plasmid backbone was amplified with oJDD6248/6249 and the two pieces were joined together using the Golden Gate BsaI-HF v2 assembly kit (NEB).

**PfPPP8 [AAs 1115-1641]^N1311,1314D-6xHis (pAM111).** The pAM101 plasmid was amplified with oJDD6545 and oJDD6546, containing modifications to change AAs 1411 and 1413 from N->D. The resulting PCR product was digested for 1 h with DpnI and transformed into *Escherichia coli*.

**ScPPT1 [AAs 209-497]-6xHis (pAM112).** The phosphatase domain of ScPPT1 (AAs 209-497[56,57]) was amplified from *S. cerevisiae* gDNA with oJDD6547/6548. The remainder of pAM101 was amplified with oJDD6549/6550 and the two pieces were joined together using the Golden Gate BsaI-HF v2 assembly kit (NEB).

**CpPPP8-[AAs 370-683]-6xHis (pAM120).** The phosphatase domain of the *C. parvum* PPP8 homolog [cgd2_1640] (AAs 370-683[56,57]) was amplified from *C. parvum* gDNA (a generous gift of the Striepen lab) with oJDD6651/6652. The remainder of pAM101 was amplified with oJDD6649/6650 and the two pieces were joined together using the Golden Gate BsaI-HF v2 assembly kit (NEB).

**ScPPT1[AAs 209-497]-MBP (pAM138).** The phosphatase domain of ScPPT1 (AAs 209-497) was amplified from pAM112 with oJDD6851/6852 and cloned into pJDD283 (containing LacI, its promoter, and the MBP), digested with EcoRI and HindIII, using the NEBuilder HiFi DNA assembly cloning kit (NEB).

**CpPPP8[AAs 370-683]-MBP (pAM139).** The phosphatase domain of CpPPP8 [cgd2_1640] (AAs 370-683[56,57]) was amplified from *C. parvum* gDNA generously provided by the Striepen lab, using primers oJDD6849/6850 and cloned into pJDD283, digested with EcoRI and HindIII, using the NEBuilder HiFi DNA assembly cloning kit (NEB).

**CpPPP8[AAs 370-683][G406D, G445R]-MBP (pAM140).** The 5' region of the phosphatase domain of CpPPP8 [cgd2_1640] was amplified from pAM139 with oJDD6849/6870 and the 3' region was amplified from pAM139 with oJDD6850/6871, containing modifications to change amino acids G406 and G445 to D and R respectively. Fragments were spliced together PCR-SOEing and cloned into pJDD283, digested with EcoRI and HindIII, using the NEBuilder HiFi DNA assembly cloning kit.

To generate all CRISPR-Cas9 guide plasmids, oligos corresponding to guides were annealed and ligated into BpiI-digested pRR216[15], which contains SpCas9 and a U6 guide cassette. For targeting PfPPP8 (pRR221), guide oligos oJDD4682/4683 were used. For targeting PfCINCH, guide oligos oJDD3927/3928 were used[15] For targeting PfBLEB, guide oligos oJDD4571/4572 were used[29]. For targeting PfMyoJ (pAM153, pAM154, pAM155), guide oligos oJDD6292/6293; oJDD6294/6295; and oJDD6296/6297 were used. For targeting PF3D7_0214700 (pRR219, 222), guide oligos oJDD4695/4696 and oJDD4693/4694 were used.

## Reagents and antibodies

Primers were obtained from IDT (Integrated DNA Technologies) or Life Technologies. The sequence of each primer used is listed in Supplementary Table 3. Restriction enzymes were obtained from New England Biolabs. Commercially available antibodies were purchased from Bio-Rad (mouse anti-V5 [MCA1360], mouse anti-Myc clone 9E10 [MCA2200GA]), Abcam (rabbit anti-6xHistidine [ab9108], rabbit anti-Histone H3 [ab1791], rabbit anti-MBP clone EPR4744 [ab119994]), Promega (rabbit anti-HaloTag [G9281]), Sigma-Aldrich (rabbit anti-Myc [C3956]), and EMD (mouse anti-CrCen [04-1624]). Secondary antibodies were obtained from ThermoFisher (goat anti-mouse IgG AlexaFluor 488 [A11029], goat anti-mouse IgG2a AlexaFluor 488 [A21131], goat anti-mouse IgG1 AlexaFluor 488 [A21121], goat anti-rabbit AlexaFluor 488 [A11008]; goat anti-mouse IgG1 AlexaFluor 555 [A21127], goat anti-mouse IgG2a AlexaFluor 555 [A21137], goat anti-rabbit AlexaFluor 555 [A21429], goat anti-rabbit AlexaFluor 647 [A21245]). Non-commercial antibodies were kindly provided by other researchers in the *P. falciparum* community: mouse anti-PfAMA1 from Robin Anders at The Walter & Eliza Hall Institute of Medical Research, rabbit anti-PfGAP45 from Julian Rayner at the Cambridge Institute for Medical

Research, mouse anti-PfMSP1, clone 1E1, from Anthony Holder at MRC National Institute for Medical Research, mouse anti-PfLDH from Michael Makler at Flow Inc. Primary rabbit antisera against PfMORN1 and PfBCP1 were previously generated in the lab[15]. The 646 nm JaneliaFluor HaloTag ligand was obtained from Promega.

## *P. falciparum* culture

The *P. falciparum* 3D7 laboratory strain, obtained from the Walter and Eliza Hall Institute (Melbourne, Australia) was the basis for all transfections and assays. Parasites were cultured in RPMI-1640 (Sigma) supplemented with 25 mM HEPES (4-(2-hydroxyethel)-1-piperazineethanesulfonic acid) (EMD Biosciences), 0.21% sodium bicarbonate (Sigma), 50 mg/l hypoxanthine (Sigma), and 0.5% Albumax II (Invitrogen). Packed RBCs were obtained from Valley Biomedical. Parasites were cultured at 37 °C with a gas mixture of 5% $CO_2$.

## *P. falciparum* transfection

To construct the lines PfPPP8-smV5[Tet], PfMyoJ-smV5, and PF3D7_0214700-smV5, 100 µg of HDR plasmid was linearized by digestion, purified, and co-transfected with 75 µg Cas9 targeting plasmid into the *P. falciparum* 3D7 strain. Lines were maintained with 500 nM ATc from the onset of transfection; 1 day post transfection, drug pressure was applied with 2.5 nM WR99210 (Jacobus Pharmaceuticals).

To construct the line PfPPP8-smV5[Tet]; PfCINCH-smMyc, 100 µg of HDR plasmid was linearized by digestion, purified, and co-transfected with 75 µg Cas9 targeting plasmid into the PfPPP8-smV5[Tet] strain. The line was maintained with 2.5 nM WR99210 and 500 nM ATc; 1 day post transfection, additional drug pressure was applied with 2.5 nM Blastocidin (Research Products International).

To construct the line PfPPP8-smV5[Tet]; PfBLEB-HaloTag, 100 µg of HDR plasmid was linearized by digestion, purified, and co-transfected with 75 µg Cas9 targeting plasmid into the PfPPP8-smV5[Tet] strain. The line was maintained with 2.5 nM WR99210 and 500 nM ATc; 1 day post transfection, additional drug pressure was applied with 2.5 nM Blastocidin (Research Products International).

To construct the line PfPPP8-smV5[Tet]; PfMyoJ-smMyc, 100 µg of HDR plasmid was linearized by digestion, purified, and co-transfected with 75 µg Cas9 targeting plasmid into the PfPPP8-smV5[Tet] strain. The line was maintained with 2.5 nM WR99210 and 500 nM ATc; 1 day post transfection, additional drug pressure was applied with 2.5 nM Blastocidin (Research Products International).

To construct the line PfPPP8-smV5[Tet]; PF3D7_0214700-smMyc, 100 µg of HDR plasmid was linearized by digestion, purified, and co-transfected with 75 µg Cas9 targeting plasmid into the PfPPP8-smV5[Tet] strain. The line was maintained with 2.5 nM WR99210 and 500 nM ATc; 1 day post transfection, additional drug pressure was applied with 2.5 µg/ml blasticidin (Research Products International).

To construct the line PfCINCH-mNeonGreen; PfPPP8-mCherry, 100 µg of HDR plasmid containing the mNeonGreen fluorescent tag fused to the 3' end of the PfCINCH ORF (pRR208) was linearized by digestion, purified, and co-transfected with 75 µg Cas9 targeting plasmid into the *P. falciparum* 3D7 strain. One day post transfection, drug pressure was applied with 2.5 nM WR99210 (Jacobus Pharmaceuticals). Following verification of PfCINCH-mNeonGreen integration, PfCINCH-mNeonGreen parasites were transfected with 100 µg HDR plasmid containing the mCherry fluorescent tag fused to the 3' end of the PfPPP8 ORF (pAM110), linearized by digestion, and 75 µg Cas9 targeting plasmid. One day post transfection, additional drug pressure was applied with 2.5 nM Blastocidin (Research Products International).

To construct the line PfCINCH-mScarlet; PfPPP8-mNeonGreen, 100 µg of HDR plasmid containing the mScarlet fluorescent tag fused to the 3' end of the PfCINCH ORF (pRR209) was linearized by digestion, purified, and co-transfected with 75 µg Cas9 targeting

plasmid into the *P. falciparum* 3D7 strain. One day post transfection, drug pressure was applied with 2.5 nM WR99210 (Jacobus Pharmaceuticals). Following verification of PfCINCH-mScarlet integration, PfCINCH-mScarlet parasites were transfected with 100 μg HDR plasmid containing the mNeonGreen fluorescent tag fused to the 3' end of the PfPPP8 ORF (pAM166), linearized by digestion, and 75 μg Cas9 targeting plasmid. One day post transfection, additional drug pressure was applied with 2.5 nM Blastocidin (Research Products International).

## PfPPP8 depletion

In all assays comparing PfPPP8-sufficient and PfPPP8-deficient parasites, PfPPP8-smV5$^{Tet}$ schizonts were purified by density centrifugation with 60% Percoll and washed twice in ATc-free complete RPMI. Schizonts were then replated into fresh erythrocytes and allowed to reinvade for 2 h. Following reinvasion, new rings were purified and synchronized by incubation with 5% (w/v) sorbitol synchronization and washed twice again in ATc-free complete RPMI. The resulting culture was then split 1:2, with half of the synchronized rings being added to RPMI with or without 500 nM ATc.

## Replication assay

Synchronized PfPPP8-smV5$^{Tet}$ (±ATC) was diluted to 0.2% parasitemia and 1% hematocrit and plated in triplicate. 100 uL of culture from each well was collected on days 1, 3, 5, and 7, washed with PBS, and resuspended in a 1:1000 dilution of SYBR green (Invitrogen) in 0.5% bovine serum albumin (BSA) in PBS. Parasites were incubated in the solution for 20 min at room temperature, then washed in 0.5% BSA in PBS and resuspended in filtered PBS. The proportion of infected cells was then determined by flow cytometry, utilizing the CellQuest Pro program for the collection and FlowJo X and GraphPad Prism 9 for analysis of results.

## Immunofluorescence

Dried blood smears were fixed with 4% paraformaldehyde for 10 min, then rinsed 3× in PBS. Following fixation and washing, parasites were permeabilized with 0.1% Triton X-100, diluted in PBS, for 10 min, and washed again in 1× PBS 3× for 3 min. Blocking solution (3% (w/v) BSA in PBS) was added to slides for 1 h at room temperature or overnight at 4 °C. Primary antibodies were diluted in blocking solution and added to slides for 1 h at room temperature or overnight at 4 °C. Following primary antibody incubation, slides were incubated 3× for 5 min in 1× PBS. Secondary antibodies, diluted in 0.5% BSA-PBS, were then added to slides for 45 min at room temperature. Slides were washed again for 5 min, 3× in 1× PBS, then incubated with Hoechst 33342 diluted at 1:5000 in PBS. After rinsing three more times with 1× PBS, coverslips were mounted with VectaShield Vibrance. For the anti-CrCen3 immunofluorescence images, a modified method was used. Parasites were allowed to settle on poly-D-lysine coated coverslips, fixed with 4% paraformaldehyde, and then processed as described above. Cells were visualized on a Zeiss LSM980 with Airyscan2 for super-resolution microscopy, or on a Nikon N-SIM S microscope. In the Zeiss LSM980 a 63× objective with a numerical aperture of 1.4 was used; in the Nikon N-SIM S, a 100× objective with a numerical aperture of 1.35 was used. Dilutions for primary antibodies are: mouse anti-V5 1:500, rabbit anti-BCP1 1:250, rabbit anti-PfMORN1 1:1000, rabbit anti-PfGAP45 1:5000, mouse anti-PfMSP1 1:500, mouse anti-PfAMA1 1:200, mouse anti-CrCen 1:500, mouse anti-Myc 1:100, rabbit anti-Myc 1:200, rabbit anti-HaloTag 1:500.

## Image processing

**Display.** For most of the Z-Stack images taken on the Zeiss LSM980 with AiryScan or Nikon N-SIM S microscopes and processed in ImageJ, each image was loaded into ImageJ and brightness and contrast were modulated solely by using the reset command (autoscale) on the brightness and contrast panel. This command restores the original brightness and contrast settings, setting the display range to the full pixel value of the image, and rendering processed images as similar as possible to native images. For certain images (listed in the relevant figures), the intensity values for each channel were matched to those of a previously "autoscaled" parasite using the SET command on FIJI.

**Measurements.** For measurements of circularity, each visible ring of the BC was traced using the FIJI freehand selection tool and the area and perimeter of each were measured. Circularity is defined as $4 \times pi \times area/perimeter^2$. Therefore, we then took the recorded area and perimeter and plugged them into a function written on RStudio to calculate circularity. A shape becomes more circular as the circularity approaches 1.

For measurements of ring diameter, each visible ring of the BC was traced using the FIJI freehand selection tool and an ellipse was fit to the freehand tracing using the set measurements and measure tools. The major diameter for each ellipse was reported. Fragmented rings that began to appear later in the PfPPP8-smV5$^{Tet}$-ATC condition were measured similarly if the break between two pieces was less than or equal to 0.4 microns and if there was no ambiguity as to what ring the fragment belonged to.

For measurements of average intensity values of fluorescent parasites, a rectangular region of interest was drawn precisely around the relevant parasite's maximum-projection image using the rectangular draw tool so each of the four sides "touched" the BC. This ROI was re-drawn at each time point so each of the four sides maintained contact with the BC; it shrank as the BC contracted and expanded as the BC expanded. The min&max gray value as well as the mean gray value were then measured using FIJI's set measurement and measure tools. The minimum gray value of each box, which stayed consistent throughout measurement, was subtracted from the mean gray value for each time point to subtract the background fluorescence. The mean gray value−min gray value for each fluorescent channel per parasite per time point was then divided by the original mean gray value−min gray value so the relative change in fluorescence could be graphed.

## Transmission electron microscopy

Synchronized late PfPPP8-smV5$^{Tet}$ schizonts, with and without ATC, were treated with 1 mM E64 at 48 hpi for 3 h. Parasites were then Percoll-purified and resuspended in fixative (2.5% paraformaldehyde, 5% glutaraldehyde, 0.06% picric acid in 0.2 M cacodylate buffer). Fixed parasites were prepared as standard for TEM and visualized on a JEOL 1200EX electron microscope.

## Live cell microscopy

For PfPPP8 depletion progression movies (Fig. 3), PfPPP8-smV5$^{Tet}$–PfBLEB-HaloTag parasites were cultured in a variation of the high parasitemia method detailed in Radfar et al.[82]. To remove ATC, a high schizontemia culture was tightly synchronized as described for washouts. Upon progression to the schizont stage, in the following cycle, 200 uL of a mixed culture of schizonts from each condition was spun down and resuspended in 50 uL of media. The JaneliaFluor 647 HaloTag ligand was added at a concentration of 800 nM and allowed to settle on one quadrant of a conconavalin A-coated ibidi/cellview glass bottom dish. After 30 min, excess/nonbound red blood cells and the HaloTag ligand were washed off with three washes of PBS, then each quadrant was filled with phenol red free RPMI, with Trolox added to a concentration of 0.5 mM to minimize bleaching and oxidative damage. Several different fields were selected based on parasite density and stage of schizont over 6.5 h, using the 647 laser at 38% laser power and 400 ms exposure, with 2 × 2 binning (500 ms exposure and no binning for the supplementary repeat). Parasites were visualized on a Nikon TiEclipse equipped with confocal imaging. For measurement, resulting

files were processed in FIJI, to produce supplementary movies, resulting files were processed in ARIVIS 4D.

For PfPPP8-PfCINCH timeline movies, PfCINCH-mNeonGreen-PfPPP8-mCherry parasites were cultured in a variation of the high parasitemia method detailed in Radfar et al.[82]. 200 uL of a mixed culture of schizonts from each condition was spun down, resuspended in 50 uL of media, and allowed to settle on one quadrant of a conconavalin A-coated ibidi/cellview glass bottom dish. After 30 min, excess/nonbound red blood cells were washed off with three washes of PBS, then each quadrant was filled with phenol red free RPMI, with Trolox added to a concentration of 0.5 mM to minimize bleaching and oxidative damage. Several different fields were selected based on parasite density and stage of schizont over 6.5 or 9 h, using the 488 laser at 13% laser power and 200 ms exposure, with 2 × 2 binning, and the 561 laser at 31% laser power, 400 ms exposure, and 2 × 2 binning. Parasites were visualized on a Nikon TiEclipse equipped with confocal imaging. For measurement, resulting files were processed in FIJI, to produce supplementary movies, resulting files were processed in ARIVIS 4D.

### Expansion microscopy

For expansion microscopy, the protocol established by Absalon et al. was followed. Synchronized PfPPP8-smV5-PfCINCH-smMyc schizonts were purified via density centrifugation with 60% Percoll. Isolated schizonts were allowed to settle on poly-D-lysine coated coverslips for 20–30 min at 37 °C. Parasites were fixed with 4% PFA for 20 min at 37 °C, washed 3× with PBS, and fixed slips were incubated with FA/AA overnight at 37 °C. Gel polymerization was performed the following morning; in ice, TEMED and APS were added to a previously made monomer solution and the coverslips were placed over a drop of the TEMED/APS/Monomer Solution mixture in a gelation chamber kept at −20 °C for at least 15 min beforehand. After 5 min of incubation on ice, the chamber was incubated at 37 °C for an hour. Post incubation, coverslips/gels were placed in 1 mL of denaturation buffer and incubated with denaturation buffer for 15 min with agitation. After gel detachment, gels were incubated for 90 more min in a 1.5 mL Eppendorf tube filled with denaturation buffer. After this incubation, denaturation buffer was removed and gels were placed in a 10 cm dish filled with ddH$_2$O, which was switched out after 30 min. After overnight incubation in ddH$_2$O, gels were washed in PBS two times for 15 min each then incubated in 3% BSA-PBS for 30 min at RT. Gels were then incubated in 1 mL of 3% BSA-PBS with primary antibodies (mouse anti-Myc, 1:100 or rabbit anti-Myc, 1:200; mouse anti-V5, 1:250) overnight at 4 °C. The next morning, gels were washed three times with 2 mL 0.1% Tween20 in PBS for 10 min at room temperature with agitation. After washes, gels were incubated in 1 mL of PBS with secondary antibodies including NHS-Ester (647 and 405, 1:100) and Hoechst (1:5000), protected from light. After 2:30, gels were washed three more times with 2 mL 0.1% PBS + 0.1% Tween20 as previously, then placed in a 10 cm dish filled with ddH$_2$O. Water was replaced after 30 min and gels were allowed to expand overnight before imaging on a Zeiss laser-scanning confocal 980 with Airyscan2.

### Immunoprecipitation

PfPPP8-smV5$^{Tet}$-PfCINCH-smMyc and parental (Pf3D7) lines were both tightly synchronized by density centrifugation with percoll followed by treatment with 5% (w/v) sorbitol after reinvasion, then expanded to 250 mL. When parasites reached 3% schizontemia, they were stalled with 2.5 uM Protein Kinase G inhibitor C1 until roughly 50 hpi. RBCs were lysed with 0.05% saponin in PBS with protease inhibitors (SigmaFast Protease Inhibitor Cocktail, Sigma). Parasite pellets were then lysed in ice-cold RIPA buffer (50 mM Tris-HCl, pH 7.5, 150 mM NaCl, 1% NP-40, 0.5% sodium deoxycholate, 0.1% sodium dodecyl sulfate) with protease inhibitors (SigmaFast Protease Inhibitor Cocktail) for 30 min on ice. Following incubation, samples were sonicated twice at 20%

amplitude for 30 s and centrifuged to remove insoluble material. The supernatant from this centrifugation was added to magnetic anti-V5 and anti-Myc beads (Pierce Anti-c-Myc magnetic beads, Product #88842, MBL Anti-V5-tag magnetic beads, Product #M167-11) and incubated overnight at 4 °C. Beads were then washed three times with RIPA + protease inhibitors, then twice with PBS plus protease inhibitors to remove detergents. They were then resuspended in 50 uL 5 mM ammonium bicarbonate and submitted to Harvard's Mass Spectrometry Core for on bead digestion and mass spectrometry analysis. Mass spectrometry results were analyzed by comparing the number of unique and total peptides and the sum intensities for each obtained protein between the experimental (PfPPP8-smV5$^{Tet}$-PfCINCH-smMyc) and control (3D7) lines. Resulting proteins were characterized as potential hits by using Microsoft Excel to filter out proteins present in the control samples and identify proteins present in both V5 and Myc samples.

### Quantitative western blotting

Parasites were isolated by lysing RBCs in 0.05% saponin in PBS with protease inhibitors (SigmaFast Protease Inhibitor Cocktail). Pelleted parasites were washed with PBS until the supernatant was clear; samples were then boiled in Laemmli buffer for 5 min (for PfMyoJ-smV5$^{Tet}$ parasites only: incubated at 60 °C for 10 min). The equivalent of 1 mL of culture at 1% schizontemia was run on a 4–20% Tris-glycine-sodium dodecyl sulfate gel and transferred to either a polyvinylidene fluoride or nitrocellulose membrane. Membranes were blocked in Licor Odyssey blocking buffer, incubated with primary antibody, and then incubated in secondary antibodies diluted in blocking buffer. Following TBST-washing, Membranes were scanned on a Licor Odyssey CLx imager system and quantified using volumetric measurement of fluorescence intensity in LiCor Image Studio 4.0. Uncropped blot images are provided in the Source Data file. Dilutions of primary antibodies were: anti-V5 1:1000, anti-MORN1 1:2000, anti-H3 1:2500, anti-LDH 1:1000, anti-Myc 1:1000. Uncropped western blot images are provided in the Source Data file.

### Statistics and reproducibility

For all single or few time point imaging experiments, three biological replicates were performed. Two biological replicates were performed for imaging via electron microscopy. For growth curve analysis, two biological replicates with three technical replicates per run were performed. For all immunoblotting experiments, three or more biological replicates were performed (four biological replicates to quantify the extent of PfPPP8 knockdown). For all live cell experiments, three biological replicates were performed. For all time course fixed-cell immunofluorescence or immunoprecipitation experiments requiring tight synchronization of parasites, close monitoring, and/or large volumes of synchronous parasite culture, two biological replicates were performed. For all recombinant protein experiments, two biological replicates with three technical replicates per biological replicate were performed.

### Data availability

The authors declare that all data supporting the findings of this study are available within the paper and its supplementary information files. Parasite lines utilized in this study are available upon request to the corresponding author. Requests for parasite strains may require an MTA per institutional guidelines prior to sharing. Genomic and protein sequences were obtained from PlasmoDB.org, VEuPathDB.org, ToxoDB.org, and PiroplasmaDB.org. Source Data are provided with this paper.

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

## Acknowledgements

The authors thank Ross Tomaino at the Taplin Mass Spectrometry Facility, Maria Ericsson at the Harvard Electron Microscopy Facility, and Paula Montero-Llopis of the Harvard Medical School MicRoN core. The study was supported by the National Institutes of Health R01 AI145941 (J.D.D.) and F31 AI157041 (A.A.M.).

## Author contributions

A.A.M.: conceptualization, methodology, validation, formal analysis, investigation, writing—original draft, and visualization. R.M.R.: conceptualization, methodology. J.D.D.: conceptualization, methodology, resources, writing—review and editing, and supervision.

## Competing interests

The authors declare no competing interests.
