## [Peer review file · Nature Communications]

REVIEWER COMMENTS

Reviewer #1 (Remarks to the Author):

In this interesting study, the authors shed some light on the mechanistically poorly characterized mechanisms of cell division of the malaria parasite Plasmodium falciparum. Specifically, they identify and characterize a PPP pseudophosphatase that localizes to the basal complex, an intriguing structure that is involved in daughter cell separation. Using a conditional expression system, they show that PPP8 is essential for the expansion and the maintenance of the integrity of the basal complex. Interestingly, evolutionary analysis showed that the protein is conserved in apicomplexa and beyond, in cells with very different modes of cell division. They further identify 2 new members of the BC with different temporal localizations.

In conclusion, this study identifies a new critical player in the essential process of merozoite formation. It also highlights the complexity of the basal complex and reveals that its composition is dynamic during the cell division process.

The work is of high quality and will be of interest to the broad readership of Nature Communication.

Major comments:

-Related to Fig 1:

A. It is clear the PPP8 highly colocalizes with BCP1 but there seem to be areas where it doesn't. I presume this means that there might be subdomains to the complex? Could the authors comment on that?

D. I do not think that the authors should put that much emphasis on AMA1 other than to say that its secretion still occurs in absence of PPP8 because it is not clear what its localization is and it detracts from the function of PPP8 at the BC. As presented, I'm not convinced that AMA1 is really mislocalized. It seems to still be secreted on the PPM. Have the authors done MSP1-AMA1 colocalization? This would provide more information if the authors really want to address the issue of where AMA1 ends up in absence of PPP8. In addition, perhaps super-res would help better resolve the localization.

F. The EM image used for the PPP8Kd does not seem representative of what is shown by IFA in D and E and supfig 2A where some merozoite formation is visible. I understand that the authors want to highlight the presence of apical organelles despite the absence of PPP8 but I think it would be better to use an image where nuclei and some defective meros are visible. Also, it would be important to use different arrows for the different apical structures.

-Line 120: As far as I'm aware, AMA1 is released on the surface of merozoites before egress from the red blood cell, not post-egress. Unless the authors refer to post-egress as the time after PVM but before RBC membrane breakdown? This should be clearly stated.

-Line 124: I do not understand what the authors mean by: but post-discharge localization was mediated by IMC and plasma membrane defects.

-Fig. 2

B-C: How is the circularity measured? Also, I'm not sure that putting an emphasis on the mean is a good way to present the data because the values do not to properly convey how

much the BC is disrupted in absence of PPP8. It leads the reader to think that it is only a subtle difference (even if it is statistically significant) whilst it is actually quite drastic. I think perhaps the variation in the circularity of individual cells, as shown in 2B, better represents that data.

D: I cannot see the labelling properly. I suggest using darker colors. In fact, these data are really cool so it would be even better to have separate panels for each marker along with the overlay, like for IFAs. This will help in better visualizing.

- SupFig3B:

There clearly seems to be less CINCH-Mic in the PPP8-deficient line by Western blot.

- Related to Fig 3:

Line 220: Could PPP8 also be important for BC contraction?

-Related to Supfig4:

Line 238: What does it mean with regards to modes of cell division that PPP8 orthologues are found in free-living protists. The evolutionary analysis deserves more discussion.

Line 249: Have the authors looked in Alphafold to compare the structures?

-Discussion:

The authors need to discuss much more the potential function of PPP8. How does it relate to the mode of cell division? The authors' group is one of the most important players in the basal complex field in Plasmodium so they are in the best position to expand and speculate.

Minor comments:

-Line 83: EstablishES

-Line 385: The authors state that MyoJ is likely essential. This is based on what? The piggyback screen? If so, it should be cited.

-SupFig 1:

A. I think it would be better to add the PCR validation here than in an appendix.

C. The authors should comment on the numerous bands that are seen by Western blot.

What is the expected size of the tagged PPP8?

-Supfig 7C:

The C1 arrest lane of the upper panel does not seem to be from the same membrane so the authors should make sure to clearly separate it. I would also add the name of the protein that is recognized by the Myc antibody on the figure, to make it easier for the reader.

-Supfig8B:

The bottom microscopy panel should be labelled PPP8-mCherry and not CINCH-mCherry.

-Supfig 9:

Looking at the figure, it seems that the MyoJ and 0214700 tagged lines are actually also Tet regulatable but no mention of this is made in the manuscript. Just to be clear, I'm not suggesting that the authors should present the detailed characterization of these lines in the current manuscript but perhaps a comment that these will be the subject of another story or something along these lines?

Reviewer #2 (Remarks to the Author):

Morano et al. have attempted to characterise the role PfPPP8 pseudophosphatase as part of the basal complex (BC) in human parasite *P. falciparum* at the asexual stage of cell proliferation called schizogony. The authors want to analyse the Basal complex in Plasmodium as this has been worked out more extensively in closely related apicomplexan parasite *Toxoplasma* and few studies in Plasmodium are starting to emerge. The authors have a review on the topic recently (Morano and Dvorin 2021).

They performed this study since they detected this phosphatase in their previous study as part of complex when they characterised another molecule Pf CINCH (Rudlaff et al 2019). They show that PfPPP8 co-localises with known BC components (such as MORN1 and BCP1), as well as being essential for asexual replication, BC integrity and uniform expansion. They have also defined it as a founding member of the PP pseudophosphatases, which is catalytically inactive. Overall, this is a nicely conducted study that characterises PPP8; however, there are a number of comments and suggestions that require addressing prior to publication

A. The major comment and experiment that will strengthen the manuscript is why the authors have not performed any IFA or expansion microscopy with Centrin when it is clearly shown that MyoJ and centrin are part of basal complex. The authors have shown the function of MyoJ and also the proteomics data but why centrin was not taken into consideration. This could give a better idea about the PPP8 and the dynamics of centrosome and basal complex. In addition where the authors show that major defects are seen after 44h of culture when segmentation of parasite starts, then it is important to know how the centrin location changes in these cells and if the cell proliferation has affected the basal complex or the centrosome dynamics. This is extremely important to get some idea of the interaction between basal complex and centrosome. This part is totally missing in every figure of IFA, live cell imaging and expansion microscopy.

B The section on PPP8 causes segmentation failure- can the author comment if the processing of AMA1 or MSP1 is altered and what is the status of these molecules in the deficient parasite with respect to sufficient PPP8 parasite and whether they have looked into it.

C. The characterisation of the PPP8 is a pseudo phosphatase etc can go at the start of the manuscript rather than after discussing all the localisation and function of PPP8 and then describing about the gene.

Few other comments are:

1) Introduction – sets the scene nicely for the BC but jumps straight into PPP8 without giving any background around PPs and what they do (or indeed what a pseudoPP is). This needs correcting. Also, the authors state that PfPPP8 is likely essential in Pf, which has also been shown in *P. berghei* (Guttery et al. 2014). This should also be highlighted.

2) Fig. 1A – the authors state that PfPPP8-smV5 co-localized with the basal complex proteins PfMORN1 and PfBCP1, but only co-localisation with PfBCP1 is shown so the localisation with MORN 1 is missing

3) Fig. 2b – please explain more clearly with all the different violin plots in the graph are (i.e. label the axes)

4) Line 176 – this statement needs data to back it up.

5) Fig. 3C – please state on the figure what the times mean (are they seconds? Minutes? Hours?)

6) Fig. 3D – please state the time units on the x axis.

7) Line 248 – any evidence to support this claim?

- 8) From over 50 proteins pulled down, I think better justification is needed as to why PF3D7_0214700 was chosen. Could the authors attempt to determine a putative homologue for this gene in non-apicomplexans?
- 9). Line 385 PfMyoJ is non essential is not shown in P.falciparum It is non-essential in P. berghei and localises to residual bodies in sporogony hence that should be given.
- 10) Fig.4d – please add in the figure what the black dots are
- 11) Fig 5C and D – can these not be combined into one bar graph?
- 12) Fig 6 C&D can the author provided a merge to show co-localisation? I appreciate that for some pictures the intensities are very bright.

Reviewer #3 (Remarks to the Author):

Summary

In this work, Morano et al. describe three new components (PfPPP8, PfMyoJ, and PF3D7_0214700) of the basal complex of Plasmodium falciparum using a combination of cutting-edge imaging techniques (super-resolution and expansion microscopy). The manuscript is well-written and easy to follow, and the authors provide a lot of quantifications of their imaging data. They thus convincingly show that the newly identified PfPPP8 is a pseudophosphatase localized to the basal complex (BC), having a critical role in the integrity and contraction of the BC and therefore in survival of the parasites. Moreover, by co-immunoprecipitation, they identified two other BC protein, PfMyoJ, and PF3D7_0214700, and provide a detailed chronology of appearance and disappearance of these proteins (and others previously identified) along the assembly, extension and contraction of the BC.

The work presented here deepens the understanding of the dynamics of the BC of Plasmodium falciparum during schizogony. This represents a critical aspect of parasite replication.

Although I am very positive about the work, I would like the authors to clarify the specific following points:

Major concerns

1. Line 94-95: it is written that PfPPP8-smV5 co-localizes with PfMORN1 and PfBCP1 in Fig 1a but only PfBCP1 is shown on the figure.
2. It is not clear what the authors mean by “post-egress” schizonts. In Fig S2a, the PfPPP8-sufficient parasites egressed after the E64 while in fig 1d, e and f, they did not. How are the parasites treated in the different experiments?
3. Fig 1f. Is it the PVM that surrounds the PfPPP8-deficient parasites or the RBC membrane? It seems very different from the PfPPP8-sufficient parasites. The authors should draw squares on the images to indicate which area was magnified. Also, the white arrows point to different structures, it would be helpful to name them (likely apical rings and rhoptries). Finally, the scale bars of the zoom-in pictures might not be correct. On the left, we see a full schizont while on the right, we have only an apical end although the scale bar is the same .
4. Fig 2b and c. It is not clear from the text which staining was used to measure the diameter of the BC. From the text, it seems that Fig 2b was done with PfCINCH while Fig 2c was done with PfBLEB. It should be mentioned in the figure legend for clarity. The unit (um) should also be added on the axis.

Lines 143-144, likely referring to Fig 2b and not 2a, mention diameters >1.5 and <0.4.

From this figure, the range seems rather between 0.2 and 1.1 but nothing >1.5. Please clarify.

Line 151: I guess Fig 2b-c should be Fig 2c.

5. Fig. S4b. A multiple alignment of the phosphatase domain including PfPPP8, CpPPP8, BdPPP8, TaPPP8, ScPPT1 but also TgBCC5 would allow to visualize more easily the conservation and differences between the catalytic residues as well as the insertions that are mainly found in the Plasmodium sequence.

Since only TgBCC5 was predicted to be active, did the authors try to express and test it in the activity assay?

Did the authors try to predict the 3D structure of PfPPP8, ScPPT1 and TgBCC5 using alpha-fold or other methods to see if the folding of the phosphatase domain is similar?

6. In the dot plot presented in Fig. 4b., the authors wrote in the legend that in magenta are the experimentally confirmed or bioinformatically predicted basal complex and inner membrane complex localized proteins without providing any references. The authors could provide these information in Table S2 by highlighting these proteins in magenta. Same for the IMC proteins in green.

7. Fig. S5a and d. What is the outline image for? The authors could indicate in this image the early and late schizonts and should explain in the figure legend what the arrows are pointing to.

8. It might be useful to add at the end a drawing recapitulating the chronology of appearance and disappearance of the BC proteins identified so far along the assembly, extension and contraction of the BC.

Minor points

Line 71. MyoJ is not a class XIV myosin but a Class VI-like (Foth et al., 2006, PMID: 16505385; Mueller et al., 2017, PMID: 29161623) or a Class XXIII (Sebé-Pedrós et al., 2014, PMID: 24443438) myosin depending of the phylogenies.

Line 277. "the BC consists of two small, attached apical rings, preceding final karyokinesis". This part of the sentence is difficult to understand, it should be rephrased for clarity.

Paragraph lines 300-315. All the citations of Supplementary Fig. 6 refer to Supplementary Fig. 7 and vice-versa. The 2 figures need to be swapped in the Supplementary file.

Line 373. The function of TgBCC5 was assessed in Roumégous et al., 2022, PMID: 36310866. Although the protein is not essential, it is involved in the intravacuolar connection between tachyzoites. This article could be cited here and maybe in the introduction line 71.

Fig. 2d and Fig. S3d. The time (x axis) could be presented in minutes rather than in hours to match the text. It would be easier to follow.

Fig. S5d. There are back boxes on the MORN1 and merge images.

Fig S8b, CIN-mCherry should be PPP8

Table S1: why is there an asterisk for the toxo ID?

Response to reviewer comments (NCOMMS-23-00279-T)

Reviewer #1 (Remarks to the Author):

In this interesting study, the authors shed some light on the mechanistically poorly characterized mechanisms of cell division of the malaria parasite *Plasmodium falciparum*. Specifically, they identify and characterize a PPP pseudophosphatase that localizes to the basal complex, an intriguing structure that is involved in daughter cell separation. Using a conditional expression system, they show that PPP8 is essential for the expansion and the maintenance of the integrity of the basal complex. Interestingly, evolutionary analysis showed that the protein is conserved in apicomplexa and beyond, in cells with very different modes of cell division. They further identify 2 new members of the BC with different temporal localizations.

In conclusion, this study identifies a new critical player in the essential process of merozoite formation. It also highlights the complexity of the basal complex and reveals that its composition is dynamic during the cell division process.

The work is of high quality and will be of interest to the broad readership of Nature Communication.

We appreciate the reviewer's time and supportive comments.

Major comments:

-Related to Fig 1:

A. It is clear the PPP8 highly colocalizes with BCP1 but there seem to be areas where it doesn't. I presume this means that there might be subdomains to the complex? Could the authors comment on that?

The anti-PfBCP1 antibody is a rabbit polyclonal antiserum raised against a peptide from PfBCP1. While extremely useful, this antibody has higher background than is seen with the anti-epitope antibodies (either V5 or myc). Therefore, with the higher background it is difficult for us to confidently describe subdomains at this resolution.

Preliminary data from our ongoing evaluation of the basal complex by ultrastructure expansion microscopy has more confidently illuminated subdomains within the complex. These studies rely upon parasite strains with two basal complex proteins tagged with different epitopes. This subdomain analysis of the basal complex is still ongoing with multiple different parasite strains. Therefore, we feel the description of the basal complex subdomains is beyond the scope of the current manuscript. We hope to have robust data to demonstrate the basal complex architecture in the future.

D. I do not think that the authors should put that much emphasis on AMA1 other than to say that its secretion still occurs in absence of PPP8 because it is not clear what its localization is and it detracts from the function of PPP8 at the BC. As presented, I'm not convinced that AMA1 is really mislocalized. It seems to still be secreted on the PPM. Have the authors done MSP1-AMA1 colocalization? This would provide more information if the authors really want to address the issue of where AMA1 ends up in absence of PPP8. In addition, perhaps super-res would help better resolve the localization.

We appreciate this comment, and it points out that we did not explain the findings clearly. We agree with the reviewer that the major point is that PfAMA1 is translocated "normally". The "abnormal" distribution reflects the primary segmentation defect. As noted by the reviewer, PfAMA1 secretion is not inhibited and the most important finding from these experiments was that the micronemes are still functional. We have changed the text to better reflect this and not emphasize the relative PfAMA1 localization as much.

F. The EM image used for the PPP8Kd does not seem representative of what is shown by IFA in D and E and supfig 2A where some merozoite formation is visible. I understand that the authors want to highlight the presence of apical organelles despite the absence of PPP8 but I think it would be better to use an image where nuclei and some defective meros are visible. Also, it would be important to use different arrows for the different apical structures.

We have replaced the TEM image in figure 1F. In addition, we now include images in the supplementary figure 2 showing multiple PfPPP8-deficient parasites observed via TEM. We have updated the main figure such that different colors of arrow are used to point out different apical organelles, and amended the legend to reflect this change

-Line 120: As far as I'm aware, AMA1 is released on the surface of merozoites before egress from the red blood cell, not post-egress. Unless the authors refer to post-egress as the time after PVM but before RBC membrane breakdown? This should be clearly stated.

This is a good point of clarification – we have replaced the wording of “post-egress” with E64-stalled throughout the section (and in the figure and the legend), described at which stage of egress E64 stalls the schizonts (after PVM rupture but before RBC membrane rupture), and altered the language describing PfAMA1's localization.

-Line 124: I do not understand what the authors mean by: but post-discharge localization was mediated by IMC and plasma membrane defects.

We apologize for the ambiguous statement. We have removed the statement as part of the rewording of the PfAMA1 translocation section.

-Fig. 2

B-C: How is the circularity measured? Also, I'm not sure that putting an emphasis on the mean is a good way to present the data because the values do not properly convey how much the BC is disrupted in absence of PPP8. It leads the reader to think that it is only a subtle difference (even if it is statistically significant) whilst it is actually quite drastic. I think perhaps the variation in the circularity of individual cells, as shown in 2B, better represents that data.

While it is true that the variance in individual parasites' circularity is more striking, the benefit of showing the difference in means (2C) is that it demonstrates a significantly lower value for circularity among PfPPP8-deficient schizonts, therefore demonstrating that PfPPP8 is important not only for basal complex rings to expand at the same rate and maintain a certain uniformity, it is also important for them to possess their circular shape. Circularity is defined as $4 \cdot \pi \cdot \text{area} / \text{perimeter}^2$ for a given shape. Therefore, we then took the recorded area and perimeter and plugged them into a function written in RStudio to calculate circularity. A shape becomes more circular as the circularity approaches 1. A description of how circularity was calculated has been added to the Methods section for the readers' benefit. In response to the reviewer's excellent suggestion, we added language *emphasizing* the increased variance of circularity measurements in PfPPP8-deficient parasites.

D: I cannot see the labelling properly. I suggest using darker colors. In fact, these data are really cool so it would be even better to have separate panels for each marker along with the overlay, like for IFAs. This will help in better visualizing.

Thank you for the suggestion. We have changed the presentation of figure 2 to allow better visualization of the findings as suggested.

- SupFig3B:

There clearly seems to be less CINCH-Mic in the PPP8-deficient line by Western blot.

In the resubmission, we have repeated this Western blot multiple times and quantified the results. A graph depicting the quantitative results is now Sup Fig 5C, and a representation of the relative amounts of PfCINCH for two replicates is now visible in Sup. Fig 5B; along with the quantitative data showing no significant difference in amount of PfCINCH. In these replicates, the amount of PfCINCH does not appear to differ by eye. Overall, upon more rigorous testing, the difference between the means of PfPPP8- sufficient and -deficient parasites' amount of PfCINCH when multiple replicates are compared and averaged seems to be minimal.

- Related to Fig 3:

Line 220: Could PPP8 also be important for BC contraction?

Yes, we certainly think PfPPP8 could be important for the contraction of the basal complex. We have added to the discussion and emphasized in the text that this is one of many possible roles for the protein.

-Related to Supfig4:

Line 238: What does it mean with regards to modes of cell division that PPP8 orthologues are found in free-living protists. The evolutionary analysis deserves more discussion.

This is a good point; to address it with the clarity it deserves we have provided more detail/analysis of this phenomenon and added it to the discussion section of the work.

Line 249: Have the authors looked in Alphafold to compare the structures?

We did look in Alphafold to compare the structures of PfPPP8, ScPPT1 and non-Plasmodium homologues. All proteins look very similar in the PPP domain; PfPPP8, however, has larger unstructured (unpredictable by AlphaFold) regions.

-Discussion:

The authors need to discuss much more the potential function of PPP8. How does it relate to the mode of cell division? The authors' group is one of the most important players in the basal complex field in Plasmodium so they are in the best position to expand and speculate.

Thank you for the vote of confidence. We appreciate this suggestion and have additional text to the discussion to directly address this important point.

Minor comments:

-Line 83: EstablishES

Thank you, we fixed it.

-Line 385: The authors state that MyoJ is likely essential. This is based on what? The piggyback screen? If so, it should be cited.

Actually, we state that PfMyoJ is likely nonessential. However, we appreciate the reminder to add the explanation for our statement and the relevant citation to the piggyBac screen, which we have added now.

-SupFig 1:

A. I think it would be better to add the PCR validation here than in an appendix.

We have moved the PCR validation panel for this line to the first supplemental figure.

C. The authors should comment on the numerous bands that are seen by Western blot. What is the expected size of the tagged PPP8?

The expected size of PfPPP8 with the SmV5 tag is ~298 kD. The lower bands are likely to be degradation products of the V5-tagged protein, which is not an uncommon consequence of using this kind of tag. While we cannot confirm this to be the case without a better primary antibody against PfPPP8 but, seeing as the lower bands become less intense in the PfPPP8-knockdown, just like the main band, it is very likely that they are degradation products/fragments of or otherwise related to PfPPP8.

-Supfig 7C:

The C1 arrest lane of the upper panel does not seem to be from the same membrane so the authors should make sure to clearly separate it. I would also add the name of the protein that is recognized by the Myc antibody on the figure, to make it easier for the reader.

This is true; the PfCINCH-smMyc last time point had to be run on a separate gel since the top part of this lane failed to transfer in the main gel. Label has also been added and adjusted.

-Supfig8B:

The bottom microscopy panel should be labelled PPP8-mCherry and not CINCH-mCherry.

Thank you for noticing this, we have corrected our mistake!

-Supfig 9:

Looking at the figure, it seems that the MyoJ and O214700 tagged lines are actually also Tet regulatable but no mention of this is made in the manuscript. Just to be clear, I'm not suggesting that the authors should present the detailed characterization of these lines in the current manuscript but perhaps a comment that these will be the subject of another story or something along these lines?

Absolutely – this is the plan! We will suggest in the discussion that the temporal localization differences suggest a potential for dimensionality to be defined for the *Plasmodium* basal complex, and the relative impact of PfMyoJ on the basal complex can be compared to that of *T. gondii*.

Reviewer #2 (Remarks to the Author):

Morano et al. have attempted to characterise the role PfPPP8 pseudophosphatase as part of the basal complex (BC) in human parasite *P. falciparum* at the asexual stage of cell proliferation called schizogony. The authors want to analyse the Basal complex in *Plasmodium* as this has been worked out more extensively in closely related apicomplexan parasite *Toxoplasma* and few studies in *Plasmodium* are starting to emerge. The authors have a review on the topic recently (Morano and Dvorin 2021).

They performed this study since they detected this phosphatase in their previous study as part of complex when they characterised another molecule Pf CINCH (Rudlaff et al 2019). They show that PfPPP8co-localises with known BC components (such as MORN1 and BCP1), as well as being essential for asexual replication, BC integrity and uniform expansion. They have also defined it as a founding member of the PP pseudophosphatases, which is catalytically inactive. Overall, this is nicely conducted study that characterises PPP8; however, there are a number of comments and suggestions that require addressing prior to publication

Again, we thank the reviewer for their time and careful reading of our manuscript.

A. The major comment and experiment that will strengthen the manuscript is why the authors have not performed any IFA or expansion microscopy with Centrin when it clearly shown that MyoJ and centrin are part of basal complex. The authors have shown the function of MyoJ and also the proteomics data but why centrin was not taken into consideration. This could give a better idea about the PPP8 and the dynamics of centrosome and basal complex. In addition where the authors show that major defects are seen after 44h of culture when segmentation of parasite starts, then it is important to know how the centrin location changes in these cells and if the cell proliferation has affected the basal complex or the centrosome dynamics. This is extremely important to get some idea of the interaction between basal complex and centrosome. This part is totally missing in every figure of IFA, live cell imaging and expansion microscopy.

We appreciate this thoughtful comment by the reviewer. *Plasmodium falciparum* has four centrins within its genome. The roles of the individual centrins have not been firmly established in any of the *Plasmodium* species during the asexual stage. Recently, it was shown by Tewari and colleagues (Roques, et al. 2018 Biology Open) that PbCen4 was dispensable for asexual replication. In this work, the other three centrins, Cen1, Cen2, and Cen3 are refractory to endogenous c-terminal tagging and are predicted to be essential for asexual replication of *Plasmodium*, presumably both *P. berghei* and *P. falciparum*. More recent work from Guizetti and colleagues (Voss, et al. 2022 bioRxiv) demonstrates that PfCen1, PfCen2, PfCen3, and PfCen4 are primarily located in the centrosome of asexual parasites when episomally expressed with GFP tags. These findings are largely congruous with our lab's experience with these genes. We have tried multiple times to endogenously epitope tag PfCen1, PfCen2, and PfCen3 without success (for many years). As a positive control, we can easily epitope tag endogenous PfCen4.

To directly answer this request, we have performed immunofluorescence using an anti-*Chlamydomonas reinhardtii* centrin antibody that likely cross-reacts with PfCen3 (Mahajan, et al. 2008 JBC). We present an analysis of PfCen3 (using the CrCen antibody) in PfPPP8-sufficient and -deficient conditions in early, middle, and late segmentation. Centrin localization does not seem to be affected in early segmentation. This lack of difference is as one would expect considering the PfPPP8-deficient phenotype is not yet apparent in early segmentation. Centrin localization does not seem to be affected in mid-segmentation schizonts deficient in PfPPP8 either, even though their basal complexes and IMC begin to become non-uniform and fragmented. In late segmentation, when severe defects are seen, centrin – at least, PfCen3 – is no longer expressed, so the significant damage to the basal complex and IMC does not appear to play a role in or impact the localization of the centrosome (at least for PfCen3).

Likely, the reviewer is discussing PfCen2 because this protein has been identified in our PfCINCH and PfPPP8 co-immunoprecipitations. Furthermore, the *T. gondii* homolog, TgCen2, is known to be part of the basal complex in this related parasite. It is important to note that TgCen2 has multiple locations in *T. gondii* (Hu 2008 PLOS Pathogens). The reviewer's request encouraged us to try again to figure out where PfCen2 localizes in asexual parasites. Because we have

been unable to endogenously epitope tag PfCen2 at its c-terminus, we made two additional parasite strains with a second copy of PfCen2 fused to smMyc. In the first strain, we integrated a second copy of PfCen2 fused to smMyc into the PfBLEB locus, replacing the coding region for PfBLEB. In this strain, the endogenous promoter of the basal complex protein PfBLEB controls its expression. We made a second parasite strain where PfCen2-smMyc is episomally expressed. This episome has the PfMCMBP promoter, which is earlier than typical basal complex proteins, to allow a broader expression timeline. The results of these studies are shown below – but neither show a definite localization to the basal complex. In the integrated parasite, shown at early segmentation, PfCen2-smMyc localizes to the centrosome (in agreement with the Voss bioRxiv paper). In mid and late segmentation, the localization of PfCen2-smMyc is more diffuse – with some centrosomal intensity, some IMC-like, and some in an unknown cytoplasmic location. Importantly, we do not see much (any?) colocalization of PfCen2 and PfPPP8, at least in these two parasite strains. Because these results are unclear and we do not know yet where PfCen2 is, we are not comfortable discussing this yet in a manuscript. We remain interested in PfCen2 but this will require additional parasite strains (and many months and potentially an additional trainee) to understand more fully.

B. The section on PPP8 causes segmentation failure- can the author comment if the processing of AMA1 or MSP1 is altered and what is the status of these molecules in the deficient parasite with respect to sufficient PPP8 parasite and whether they have looked into it.

We have not looked into the impact on the processing of PfAMA1 and PfMSP1 of PfPPP8-knockdown largely because both proteins still localize to the plasma membrane immediately preceding egress / following PVM permeabilization. Morphological defects in IMC formation lead to improperly ‘packaged’ merozoites, but they are still capable of egress. If PfAMA1 and PfMSP1 did not localize to the plasma membrane following the conclusion of segmentation / immediately preceding egress, we would have investigated whether PfPPP8 deficiency impacts their processing, but since PfPPP8-deficient parasites possess properly formed apical organelles, and micronemal proteins are capable of translocation, we do not expect processing of either protein to be tied to PfPPP8’s presence or function within the basal complex.

C. The characterisation of the PPP8 is a pseudo phosphatase etc can go at the start of the manuscript rather than after discussing all the localisation and function of PPP8 and then describing about the gene.

While we appreciate the suggestion, we prefer to have the pseudophosphatase / mechanistic discussion placed after the discussion of PfPPP8’s essentiality and the consequences of its knockdown, as it follows the flow of our research – we only became interested in the structure of and mechanism of PfPPP8’s function because of its essentiality, and the dramatic phenotype we saw upon removal of ATC from these parasites.

Few other comments are:

1) Introduction – sets the scene nicely for the BC but jumps straight into PPP8 without giving any background around PPs and what they do (or indeed what a pseudoPP is). This needs correcting. Also, the authors state that PfPPP8 is likely essential in Pf, which has also been shown in *P. berghei* (Guttery et al. 2014). This should also be highlighted.

We thank you for reminding us of this citation, and added it to the beginning of the results section where we discuss why PfPPP8 was predicted to be essential, citing its essentiality in *P. berghei* as evidence for its putative essentiality in *P. falciparum*. In addition, we have amended the introduction to include the definition and a few broad functions of pseudophosphatases, so the reader has more information about this group of pseudoenzymes going in.

2) Fig. 1A – the authors state that PfPPP8-smV5 co-localized with the basal complex proteins PfMORN1 and PfBCP1, but only co-localisation with PfBCP1 is shown so the localisation with MORN 1 is missing –

Thank you for this reminder, we have added colocalization of PfPPP8 with PfMORN1 in figure 1.

3) Fig. 2b – please explain more clearly with all the different violin plots in the graph are (i.e. label the axes) –

We added more description on why we chose to measure circularity, what the axes represent, and what each violin plot means in the text. We have also made clear in the legend to explain that each violin represents a set of circularity values recorded for one parasite.

4) Line 176 – this statement needs data to back it up.

We added a call to figure 3 where we demonstrate this qualitatively and quantitatively in two different parasite lines.

5) Fig. 3C – please state on the figure what the times mean (are they seconds? Minutes? Hours?)

This has been added to the legend and to all legends with live cell imaging data; thank you!

6) Fig. 3D – please state the time units on the x axis.

This has been added to the figure.

7) Line 248 – any evidence to support this claim?

We have mentioned the slight evidence we have regarding additional differences between these and functional phosphatases and altered the language around this discussion.

8) From over 50 proteins pulled down, I think better justification is needed as to why PF3D7_0214700 was chosen. Could the authors attempt to determine a putative homologue for this gene in non-apicomplexans?

We have modified the text to explain our choice of this protein better: Pf3D7_0214700 was selected as a protein to examine because it was both completely uncharacterized, thus being quite distinct from non-apicomplexan proteins with defined domains and motifs, and *because* of its uniqueness to *Plasmodium* spp. We have elaborated on other attractive properties of this protein, namely its small size which will more readily allow us to produce recombinant proteins in the event of *in vitro* experimentations, and on the importance of selecting both proteins with homologs in *T. gondii* and other apicomplexans, and those without. Understanding how the basal complex functions in schizogony separately and differently from in endodyogeny is an important question we have begun to ask since proteins such as PfMORN1, with an essential homolog in *T. gondii*, have been proven dispensable in asexual *Plasmodium* replication.

9). Line 385 PfMyoJ is non essential is not shown in *P. falciparum* It is non-essential in *P. berghei* and localises to residual bodies in sporogony hence that should be given. –

We appreciate the reviewer's suggestion and apologize for missing this citation. We have added it and mentioned that it was shown to be nonessential in *P. berghei* and only predicted to be nonessential in *P. falciparum* due to the mutagenesis screen.

10) Fig.4d – please add in the figure what the black dots are

We have altered the legend to define the black dots for figure 4B (not 4d) in the legend.

11) Fig 5C and D – can these not be combined into one bar graph?

They technically could but we prefer that they stay represented as two different bar graphs to illustrate the striking similarity in the temporal localizations of these two proteins.

12) Fig 6 C&D can the author provided a merge to show co-localisation? I appreciate that for some pictures the intensities are very bright.

Thank you for this suggestion, we have added merged images to this figure.

Reviewer #3 (Remarks to the Author):

Summary

In this work, Morano et al. describe three new components (PfPPP8, PfMyoJ, and PF3D7_0214700) of the basal complex of *Plasmodium falciparum* using a combination of cutting-edge imaging techniques (super-resolution and expansion microscopy).

The manuscript is well-written and easy to follow, and the authors provide a lot of quantifications of their imaging data. They thus convincingly show that the newly identified PfPPP8 is a pseudophosphatase localized to the basal complex (BC), having a critical role in the integrity and contraction of the BC and therefore in survival of the parasites. Moreover, by co-immunoprecipitation, they identified two other BC protein, PfMyoJ, and PF3D7_0214700, and provide a detailed chronology of appearance and disappearance of these proteins (and others previously identified) along the assembly, extension and contraction of the BC.

The work presented here deepens the understanding of the dynamics of the BC of *Plasmodium falciparum* during schizogony. This represents a critical aspect of parasite replication.

Although I am very positive about the work, I would like the authors to clarify the specific following points:

Major concerns

1. Line 94-95: it is written that PfPPP8-smV5 co-localizes with PfMORN1 and PfBCP1 in Fig 1a but only PfBCP1 is shown on the figure.

We appreciate this suggestion by this and the other reviewers. We have added images of PfPPP8 colocalizing with PfMORN1 as well in figure 1.

2. It is not clear what the authors mean by “post-egress” schizonts. In Fig S2a, the PfPPP8-sufficient parasites egressed after the E64 while in fig 1d, e and f, they did not. How are the parasites treated in the different experiments?

We have modified the text to explain the stage of egress represented by E64-stalled parasites and removed the vague term “post-egress” from the manuscript and replaced it with “E64-stalled”. We also describe in the text that E64 stalls the schizonts after PVM rupture but before RBC membrane rupture, so the timing of the parasites in question is more precisely described. All E64-stalled parasites were treated the same way – a common phenotype of E64-stalled parasites is that they appear as a collection of merozoites in the RBC membrane, which has not ruptured, however there can be artificial / manual rupture of the RBCM during the smearing process used to generate IFAs.

3. Fig 1f. Is it the PVM that surrounds the PfPPP8-deficient parasites or the RBC membrane? It seems very different from the PfPPP8-sufficient parasites.-

For both parasites, it is the RBC membrane surrounding the merozoites, although the PVM may remain to some degree in the PfPPP8-sufficient parasite as it may be slightly earlier in the segmentation process (perfect synchronization is never possible). It was an artifact of preparation that the PPP8-KD that the RBC membrane ruptured enough to clear the cytoplasm in the PfPPP8-deficient parasites; while this image had the best view of intact apical organelles in PfPPP8-deficient parasites an additional supplementary figure has been created composed of multiple EM images of PfPPP8 deficient and sufficient schizonts to showcase the multiplicity of phenotypes present upon PfPPP8-knockdown, and illustrating that the “cleared membrane” is not unique to PfPPP8-deficient parasites. In response to this comment and others from the reviewer’s above, we have changed the PfPPP-deficient TEM image for figure 1f.

The authors should draw squares on the images to indicate which area was magnified. Also, the white arrows point to different structures, it would be helpful to name them (likely apical rings and rhoptries). Finally, the scale bars of the zoom-in pictures might not be correct. On the left, we see a full schizont while on the right, we have only an apical end although the scale bar is the same .

The arrows have been color-coded to correspond to different apical organelles (namely APR and rhoptry) and pointed out in the legend. We have added squares to indicate where the magnification was taken. Finally, while the scale bars are the same length, that length corresponds to 500 nm in the full schizont and 100 nm on the apical end image. We have emphasized this distinction in the legend and the figure for clarity.

4. Fig 2b and c. It is not clear from the text which staining was used to measure the diameter of the BC. From the text, it seems that Fig 2b was done with PfcINCH while Fig 2c was done with PfbLEB. It should be mentioned in the figure legend for clarity. The unit (um) should also be added on the axis.

We very much appreciate this request for improved clarity. These are measurements of circularity not diameter and do not have units – and are based on PfcINCH staining. In our initial submission, this section was unclear and contained some sentences should have been moved to a later part. The section describing figure 2 (and supplementary figure 4, which shows a similar, though unquantified, phenotype with PfbLEB-HaloTag) discusses our discovery of qualitative differences between the basal complexes of PfPPP8-sufficient and -deficient parasites, which we first quantify using the measure of circularity. In the resubmission, we have improved the clarity of this section, removed the initial mention of diameters, added extra information about circularity to the text, and clarified the figure legend.

Lines 143-144, likely referring to Fig 2b and not 2a, mention diameters >1.5 and <0.4 . From this figure, the range seems rather between 0.2 and 1.1 but nothing >1.5 . Please clarify.

Again, we thank the review for this request. In the original submission, this text was incorrectly placed – data regarding differences in diameter is all in figure 3 (and supplementary fig 5). Circularity, by definition, cannot be greater than 1, so we have altered the graphs in figure 2 to help make this clear and separate these measurements from those of diameter in the following section. We have now strictly delineated information about circularity and diameter in separate sections to reduce confusion about which measurement is being taken and clarified this on the appropriate figure legends as well.

Line 151: I guess Fig 2b-c should be Fig 2c.

We apologize, but this statement is unclear. The discussion of circularity includes both Fig 2b and c.

5. Fig. S4b. A multiple alignment of the phosphatase domain including PfPPP8, CpPPP8, BdPPP8, TaPPP8, ScPPT1 but also TgBCC5 would allow to visualize more easily the conservation and differences between the catalytic residues as well as the insertions that are mainly found in the Plasmodium sequence. –

We agree that this is an excellent idea. We have replaced Fig. S4b (this is now supplementary figure 6) with the multiple alignment of these six proteins' phosphatase domains. We also added, as an additional supplementary file (supplementary file 1), an alignment of the phosphatase domains of every protein mentioned in supplementary table 1.

Since only TgBCC5 was predicted to be active, did the authors try to express and test it in the activity assay?

We have altered the text to remove the line about predicted activity – TgBCC5 was not the only homolog predicted to be active; all the *Cryptosporidium* homologs were as well, despite the presence of likely inactivating mutations. As we demonstrated, both the wild type and back-mutated *C. parvum* PPP8 homolog's phosphatase domain were inactive compared to ScPPT1. Evidently, the algorithm used was not able to pick up on even more obvious changes that would render this type of phosphatase inactive, so long as they were not as severe as those in PfPPP8's phosphatase domain. Therefore, we removed lines about predicted activity/inactivity as they are not reflective of our data or the sequences of extant proteins. Back-mutated *C. parvum* PPP8 is strikingly similar to TgBCC5 in terms of sequence, with a similar number of insertions between motifs within the domain. We also recombinantly produced and tested the activity of the *V. brassicaformis* homolog's phosphatase domain, which is even closer to that of ScPPT1 than either TgBCC5 or the altered CpPPP8; we did not include this data since the *V. brassicaformis* phosphatase domain was also inactive and it seemed redundant. Thus, we think that despite the algorithm's prediction of activity for all three of

these phosphatases related to PfPPP8, there are likely to be other less highlighted/less obvious differences that render them inactive.

Did the authors try to predict the 3D structure of PfPPP8, ScPPT1 and TgBCC5 using alpha-fold or other methods to see if the folding of the phosphatase domain is similar?

We did – just looking at Alphafold, the *Cryptosporidium* homologs look most similar to ScPPT1 and hPP5c (a human confirmed experimentally active serine/threonine protein phosphatase). PfPPP8 has unstructured / poorly predicted domains surrounding the phosphatase domain. All the apicomplexan homologs have phosphatase domains that look a little bit more ‘spread out’ than those of the experimentally confirmed phosphatases, with the most extreme example being PfPPP8 (the Plasmodium species have added multiple amino acids between core motifs of the phosphatase domain). However, we are not experts on this so will refrain from making definitive statements about how well these translate .

6. In the dot plot presented in Fig. 4b., the authors wrote in the legend that in magenta are the experimentally confirmed or bioinformatically predicted basal complex and inner membrane complex localized proteins without providing any references. The authors could provide these information in Table S2 by highlighting these proteins in magenta. Same for the IMC proteins in green. –

We have updated the color scheme of Table S2. The green proteins are not IMC proteins but are PF3D7_0214700 and PfMyoJ. We have described this more clearly in the updated legend.

7. Fig. S5a and d. What is the outline image for? The authors could indicate in this image the early and late schizonts and should explain in the figure legend what the arrows are pointing to. –

This is an excellent point. Our initial description was not clear. We have increased figure clarity by explaining the purpose of the ‘outline images’ (it delineates the borders of each parasite, emphasizing that we are looking at two different parasites at different stages of schizogony in each single image) and added two different types of arrows to differentially indicate “early” (pre-contraction) and “late” (contraction has begun) schizonts.

8. It might be useful to add at the end a drawing recapitulating the chronology of appearance and disappearance of the BC proteins identified so far along the assembly, extension and contraction of the BC. –

We appreciate the suggestion and have added this as supplementary figure 13.

Minor points

Line 71. MyoJ is not a class XIV myosin but a Class VI-like (Foth et al., 2006, PMID: 16505385; Mueller et al., 2017, PMID: 29161623) or a Class XXIII (Sebé-Pedrós et al., 2014, PMID: 24443438) myosin depending of the phylogenies.

Thank you for the correction, this has been fixed and a citation added.

Line 277. “the BC consists of two small, attached apical rings, preceding final karyokinesis”. This part of the sentence is difficult to understand, it should be rephrased for clarity.

We have changed the text to make it more clear.

Paragraph lines 300-315. All the citations of Supplementary Fig. 6 refer to Supplementary Fig. 7 and vice-versa. The 2 figures need to be swapped in the Supplementary file.

We have updated / altered the numeration of all supplementary figures; this issue has been resolved in the process.

Line 373. The function of TgBCC5 was assessed in Roumégous et al., 2022, PMID: 36310866. Although the protein is not essential, it is involved in the intravacuolar connection between tachyzoites. This article could be cited here and maybe in the introduction line 71.

Thank you so much for alerting us to the existence of this paper, we are embarrassed that we didn’t know about this publication and have edited our discussion to include a comparison/contrast of PfPPP8 and TgBCC5’s respective spatial and temporal localizations as well as knockdown phenotypes. This has strengthened the discussion immensely.

Fig. 2d and Fig. S3d. The time (x axis) could be presented in minutes rather than in hours to match the text. It would be easier to follow.

Since every other timeline for imaging is in the hours:minutes format, we instead changed the text so that the time points are described in terms of hours:minutes to both maintain clarity and ensure all the graphs have the same time scale.

Fig. S5d. There are black boxes on the MORN1 and merge images.

We appreciate the close attention to detail. The black boxes are covering the original scale bars from Zen (the software used to run our microscope) which became pixelated when the image was enlarged. Therefore, for improved crispness of the text, we covered the original with a black box and replaced the scale bar with a vector image.

Fig S8b, CIN-mCherry should be PPP8

Thank you for catching this mistake, we have remedied it!

Table S1: why is there an asterisk for the toxo ID?

This was a typo, thank you for catching it; it has since been deleted.

REVIEWERS' COMMENTS

Reviewer #1 (Remarks to the Author):

The authors have satisfactorily responded to my comments.

I recommend the paper for publication.

Congratulations on a great piece of work.

Reviewer #2 (Remarks to the Author):

The authors have addressed most of the comments. It is still surprising that they could not get centrin localisation in their mutant. But their effort to tag centrin and inability to precisely get the result is appreciated.

One comment was on the MSP1-AMA processing and location. It seems that MSP1 localisation was affected. So it would have been nice to see if the processing of MSP1 is compromised in these mutants. However I take the argument of the authors and hence may not be required.

Reviewer #3 (Remarks to the Author):

As mentioned in my previous review, the work presented here deepens the understanding of the dynamics of the BC of *Plasmodium falciparum* during schizogony. This represents a critical aspect of parasite replication.

The authors have been very responsive to reviewer comment, addressed all the concerns and clarified grey areas, making the paper even stronger.

This beautiful study is of high quality and will be of interest to the broad readership of *Nature Communication*.

Response to Reviewers

Reviewer #1 (Remarks to the Author):

The authors have satisfactorily responded to my comments.

I recommend the paper for publication.

Congratulations on a great piece of work.

We appreciate the supportive comments.

Reviewer #2 (Remarks to the Author):

The authors have addressed most of the comments. It is still surprising that they could not get centrin localisation in their mutant. But their effort to tag centrin and inability to precisely get the result is appreciated.

We are surprised as well that the localization of PfCen2 was so difficult to pin down. We are continuing to investigate this with additional parasite strains. However, if PfCen2 is an important part of the basal complex in *Plasmodium falciparum*, we will do our best to find it and report it in the future.

One comment was on the MSP1-AMA processing and location. It seems that MSP1 localisation was affected. So it would have been nice to see if the processing of MSP1 is compromised in these mutants. However I take the argument of the authors and hence may not be required.

This is an important point. However, we state that MSP1 trafficking is not abnormal. Rather, the differential localization of MSP1 is secondary to the segmentation defect caused by disruption of the basal complex. In other words, the MSP1 is normally trafficked to the abnormally formed plasma membrane. Furthermore, the normal translocation of PfAMA1 is additional data that proteins in the secretory system are largely trafficked (and presumably processed) normally. Therefore, we have not further amended the text to address this issue any further.

Reviewer #3 (Remarks to the Author):

As mentioned in my previous review, the work presented here deepens the understanding of the dynamics of the BC of *Plasmodium falciparum* during schizogony. This represents a critical aspect of parasite replication. The authors have been very responsive to reviewer comment, addressed all the concerns and clarified grey areas, making the paper even stronger.

This beautiful study is of high quality and will be of interest to the broad readership of Nature Communication.

Thank you for your support.